# Representation of the spatio-temporal narrative of *The Tale of Li Wa*李娃传

**Zhaoyi Ma** ⊚, **Jie He**‡*, **Shuaishuai Liu**‡

School of Architecture, Tianjin University, Tianjin, China

⊚ These authors contributed equally to this work.
‡ These authors also contributed equally to this work.
* janushe@tju.edu.cn

**Data Availability Statement:** All relevant data are within the manuscript and its Supporting Information files. All files are also available from the PROTOCOL (http://dx.doi.org/10.17504/protocols.io.bdyni7ve) and the github (https://github.com/aayi/The-Tale-of-Li-Wa).

## Abstract

This paper posits a framework of digital models integrating spatial narrative theories to represent the narrative and its experience of a Chinese classical novel, *The Tale of Li Wa*, which has been diversely interpreted by literary critics and historians for approximately 900 years. In the proposed framework of "Time–Space–time–Space," the spatio–temporal information derived from the text of the novel and its author is extracted and fused to map the instantaneous spatial pattern perceived by readers in the flow of read time, which helps contemporary readers re-understand the classical narrative and its context through a multi-possibilities, integrated, and in-depth approach. The paper presents one possible interpretation of the novel in the framework organized by the authors, illustrating the theme of growth with respect to the male protagonist in the capital Chang'an via the "what-how-why" model.

## Introduction

Narrative is the basic impulse inherent in human beings and uses narrative structures to convey the experiences of providing guidance or aiding a comprehension of the problematic world [1]. Historical narratives of political events and historic figures have a long history in China that began in the fifth century BC. Ancient Chinese scholars began applying their narrative experiences into fictional novels illustrating daily life from the early first century onward, and drama was added later [2]. The Tang dynasty (618–907 AD) was one of the peak periods of Chinese culture, and when Chinese classical novels gradually matured from the shadow of historical biography and supernatural novels. The so-called *Tang Tales* (唐传奇) novels have always been regarded as having great literary and social historical value, which is largely attributed to the unique social and cultural contexts of the Tang dynasty behind their narratives. The contexts include the prosperity and complexity of metropolitan areas; the fashion of submitting candidates' portfolio, called Xingjuan (行卷) in Chinese, to a senior officer before the Imperial Examinations; the dissemination of Taoist stories and fairytales; the expansion of the townspeople stratum; and the complexity of social life [3].

One of the most famous Tang Tales, *The Tale of Li Wa* (李娃传), which was written by literati member Mr. Bai Xingjian (白行简), who lived from 776 to 826 A.D, tells a twisted love story between a tribute student (an examinee from the province), Student Zheng (荥阳生) and

**Funding:** The present research is supported by the National Natural Science Foundation of China (Grant No. 51978448), Data Mining and GIS Platform for Landscape Recognition in Frontier Poetry of Quan Tangshi: A Digital Humanities Approach; and by the Program of Introducing Talents of Discipline to Universities (Project No. B13011).

**Competing interests:** The authors have declared that no competing interests exist.

a courtesan named Li Wa (李娃) during the Tianbao (天宝) Period (742–756 AD) (Fig 1). This story is a model of the "scholar and beauty (才子佳人)" literature and resulted in refurbished versions of dramas and narratives in succeeding dynasties. The story begins when a

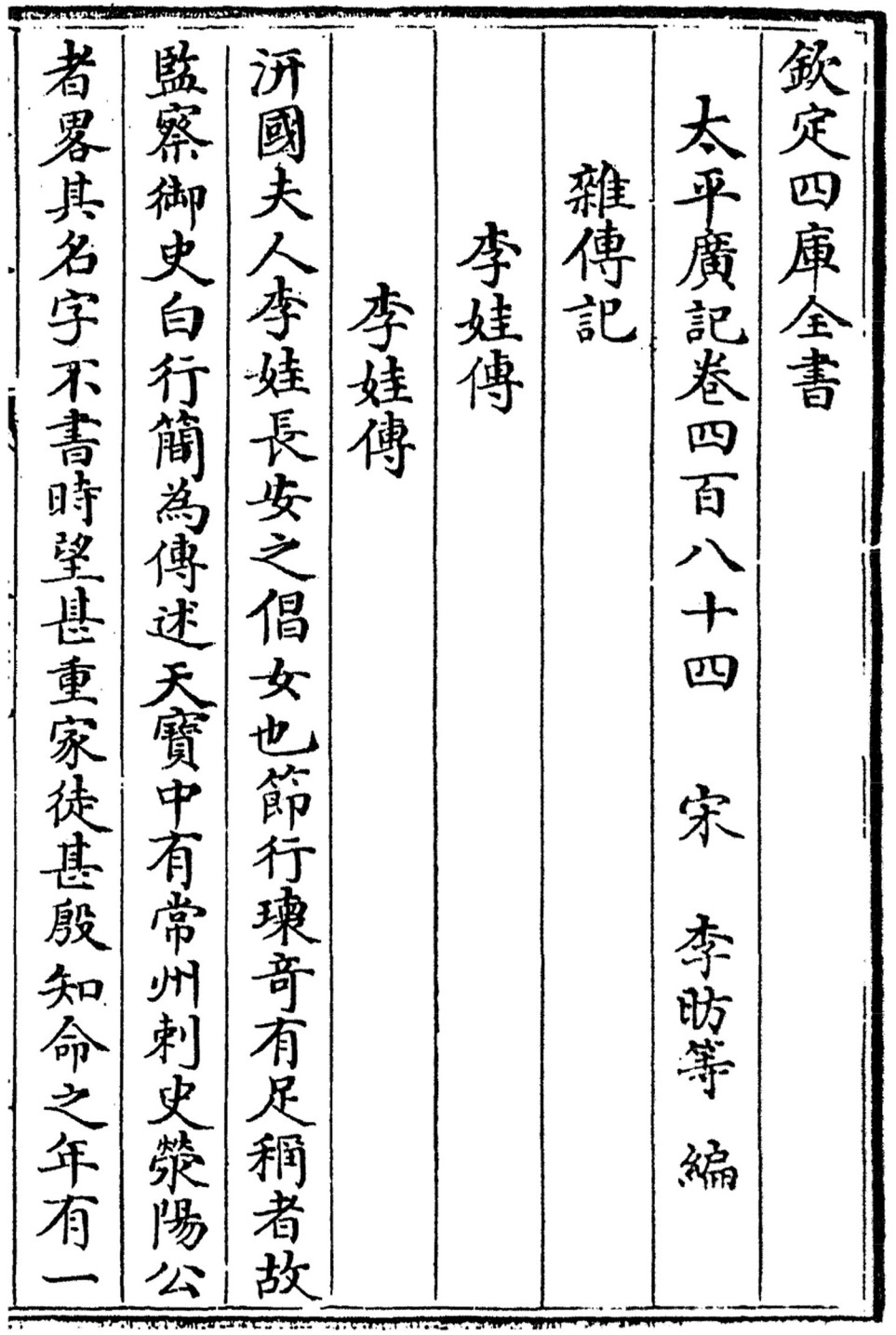

**Fig 1. Electronically scanned version of *The Tale of Li Wa* in *Complete Library in Four Sections* (四库全书).**

young tribute student enters Chang'an city (长安), the capital of the Tang Empire, where he is fascinated by the courtesan Li Wa and subsequently lives with her. More than a year later, Li Wa and her madam (李娃姥), both of whom have exhausted Zheng's fortune, abandon him. Penniless and ill, he starts working as an elegy singer in the city and then wins a public singing competition on the capital's central boulevard while his father is visiting the capital for business. His father recognizes Zheng and beats him nearly to death in anger, following which Zheng becomes a street beggar. Finally, Li Wa discovers Zheng and nurses him back to physical and spiritual health. He returns to his abandoned career, successfully passes the Imperial Examination, earns his degree, begins his bureaucratic career, and reconciles with his father. With the support of his father, Li Wa eventually assumes the role of Zheng's virtuous wife.

As is typical for Tang Tales, the novel has been well discussed and analyzed by scholars in the past, since approximately 900 years ago, in terms of the writing background, ideological theme, characterizations, artistic characteristics, cultural studies, gender studies, transmission, and acceptance [4]. However, relevant discussions require a profound academic foundation or substantial background information, particularly of classical narrative literature in the context of ancient places and lives. Moreover, the obscure and abstruse expressions of classical Chinese have led to the following: modern readers, excluding professional researchers and literary enthusiasts, more accustomed to fragmented reading and picture reading, may find it difficult to fully understand the experiences narrated in *The Tale of Li Wa* in depth, although now the novel has even been included in some Chinese high school textbooks. The innate contextual difference between the present and the past has raised the crucial topic of comprehending the classical narratives and their contexts in a more integral, in-depth, and novel manner.

In contemporary mainstream narrative theory, which emphasizes both literary works and context, the close relationship between time and narrative has been widely recognized for linear characteristics of language [5]. However, compared to the dimension of time, space has received considerably less attention. Not until the further development of novels and theories on the novel in the twentieth century were the topics of space and structure considered [6]. Nevertheless, space is a basic feature of ancient Chinese narratives. The space–time format wherein a time interval is used to replace a space interval and subsequently rearranged is a typical Chinese tradition in narratives and is defined by a Chinese term "narrative" (序事), appearing in Qin dynasty (i.e., second century BC) [2].

Heavily influenced by early historical narratives and widely disseminated among the literati in the Tang dynasty, *The Tale of Li Wa* and the other Tang Tales have a remarkable feature of spatio–temporal narrative that is primarily based on some authenticity of that time. The spatio–temporal narrative and the real spatial context behind *The Tale of Li Wa* have been considerably and diversely interpreted in many studies. Typical researches can be found in Zhu Mingqiu (朱明秋)'s analysis of the complexity and overlap of the plots from a perspective of mathematical criticism [7]; Cheng Guobin (程国斌)'s discussion on the concept of the family's status, emphasizing the successful marriage [8]; and Seo Tatsuhiko (妹尾達彦)'s study on the relationship between text plots and Chang'an wards (坊里) [9]. Most of the aforementioned academic studies are not much easier to read than the classical novel itself. Moreover, because of the limitations of perspective and the opacity of subjective feelings, it is difficult for readers to synthesize their own experience with previous researches. Rather than the artistic skills and themes of literary critics or local knowledge emphasized by historical geographers, an integrated and exploratory approach toward collecting and representing its narrative from the dimensions of space and time helps contemporary individuals lacking the relevant context organize and profoundly understand the experience and knowledge of such classical Chinese novels in a synchronic and diachronic manner.

Our discussion of the spatio–temporal narrative refers to three theories of spatial narrative: Mikhail Bakhtin's chronotope (time–space), which focuses on the connection between text and social historical context [10]; Gabriel Zoran's topographical–chronotopic–textual space, which emphasizes that the fictional literature space is a reconstruction process involving readers [11]; and WJT Mitchell's spatial metaphor, which focuses on "a spatial apprehension of the work as a system for generating meaning" [12]. The integration of these three theories constructs the context, spatio–temporal process, and the metaphorical significance of narratives from the perspective of readers, which can be deftly integrated with semantic analysis and two-dimensional or three-dimensional visualization through digital technology. Contemporary readers are in a digital age wherein the dialog between humanities data and computation moves from text digitization to text structuring, and then to text semanticizing. Representational and interpretive practices in relation to literary texts through quantitative analysis and transmedia argumentation can be traced in Franco Moretti's notion on "distant reading" in 2000 and his practice of literary cartography integrating "spatial criticism" in 1999 [13,14]. Mapping a linear text such as *The Tale of Li Wa* to a visual semantic graph allows the story to be reorganized and interpreted through new methods of representation into applicable knowledge for different readers.

This representation of the spatio–temporal narrative is not based on specific content or assumptions but emphasizes the process of reorganizing the narrative of classical Chinese literature. The present research can be considered as a comprehensive exploration toward narratives, experiences and geographical spaces [15] under an open framework. It is an opaque groping toward the interpretation of classical Chinese literature from a non-professional perspective of literature. The most crucial task while reading, according to Mitchell's narrative theory, is to excavate or provisionally impose "spatial patterns on the temporal flow of literature" [11] (p.552-553). A continuous and unpredictable reading process helps readers understand the rich connotations conveyed by *The Tale of Li Wa*, including the physical texts and the construction of the world of meanings through our deep, experiential, and visualized representation of the classical narrative in the framework of the chronotopes involving the author, the story, and the readers.

## Materials and methods

http://dx.doi.org/10.17504/protocols.io.bdyni7ve.[PROTOCOL DOI]

Reading comprehension is a complex process that involves interaction between texts, readers, and authors that share the same frame among chronotopes. The term "chronotope" was adapted by Bakhtin based on Einstein's theory of relativity and refers to a spatio–temporal literary unity. The three chronotopes of different aspects determine the artistic unity of the relationship between narratives and reality, and a contained value factor can be sorted and decomposed through an abstract analysis of chronotopes [10] in text, maps, and other materials. The chronotope of the text analyzed in this paper is a reconstructed Chang'an under dimensions of story time and story space, which can be represented by the structural semantics of texts; the reader's chronotope refers to the contemporary world in the read time (defined as the sequence of phrases called judou (句读) in traditional Chinese), illustrated by the reading mappings of readers. By contrast, the author's chronotope points to the physical aspects of Chang'an city from the author Bai Xingjian's real experience and can be interpreted from archaeological maps and his biographical records. These chronotopes, which can be extracted into different abstract variables to represent the unity of the reader's time and "author's" space, are constantly compared to each other in the logical loop starting from temporal, going to spatio–temporal, and coming to spatial. Reading comprehension deepens in this integration path (Fig 2).

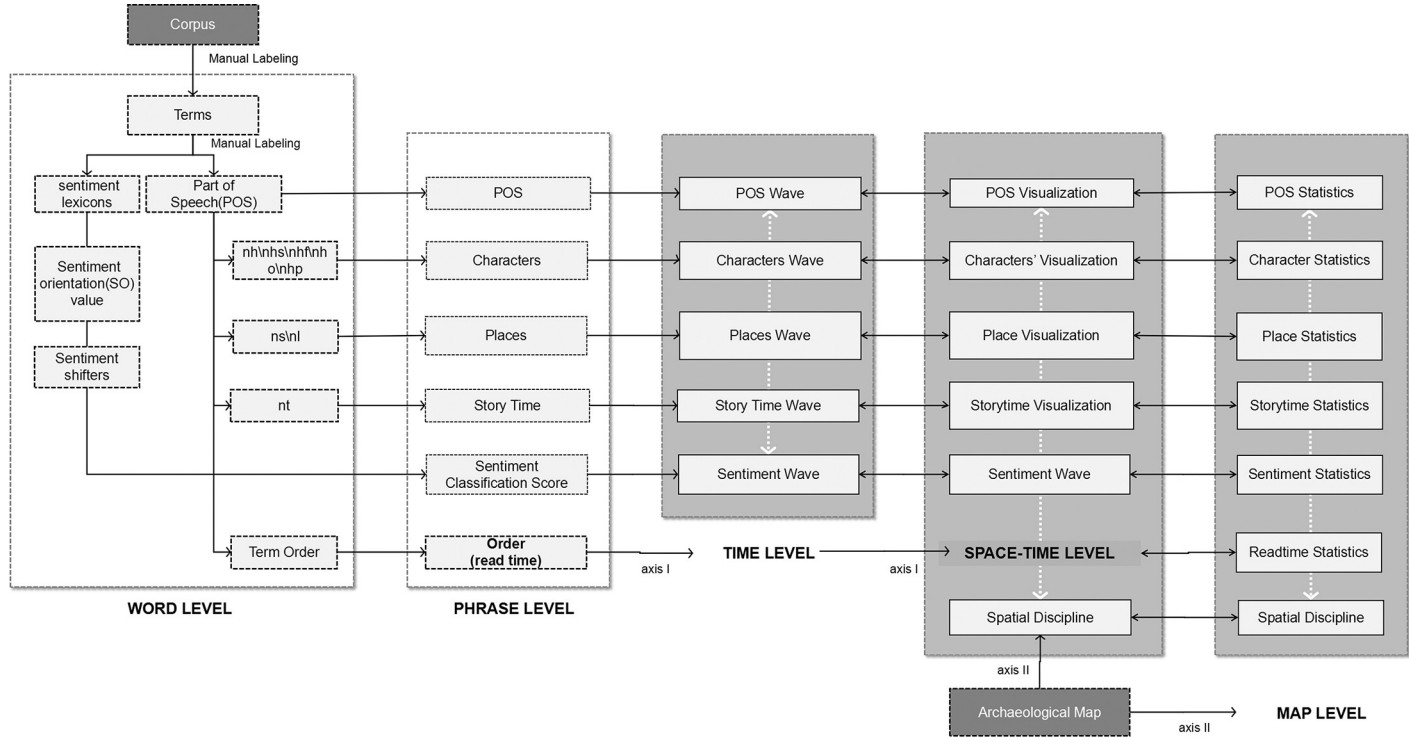

**Fig 2.** Flows (black arrows) of variables and comparisons (white arrows) among variables in the logical loop of time–space–time–space.

## Word-level analysis

Word-level analysis comprises structuring and semanticizing physical texts. *The Tale of Li Wa* is written in Middle Chinese (languages used in China from the fifth to the tenth century) with no punctuation, and it is included in the *Records of the Taiping Era* (太平广记*)*, which was edited in 978 AD. The polysemy of Middle Chinese [16] and the lack of specific natural language processing tools caused difficulties in structuring the novel's texts. Because the novel has less than 4,000 Chinese characters as per its *Complete Library in Four Sections (*四库全书*)* version (see S1 File), a text database is manually created that includes *terms* (unigram) [17], *parts of speech* (POS) [18] (p.49), *sentiment orientations* (SO) value, and *sentiment shifters* [18] (p.59) (see S1 Table). Further, because there are no ready-made sentiment lexicons, the SO value is manually assigned in two rounds as follows: each positive sentiment expression in the novel, such as laugh (欢笑) (v.) and magnificent (瑰奇) (a.), is given an SO value of +1 (172 in total); each negative sentiment expression such as whimper (呜咽) (v.) and poor (贫窭) (a.) is assigned a SO value of −1 (177 in total). The conformance percentage of two rounds of value assignment is more than 80%. Moreover, preliminary statistics indicate that Middle Chinese may not conform to the current general linguistic hypothesis that sentiment words have context-dependent orientations [18] (p.193). The tested SO value of a term is determined by its cosine similarity (based on the word2vec model [19]) with all other terms with manually-collected SO value, and the distribution of all terms' tested SO value is greatly inconsistent with that of manual assignment.

## Phrase-level analysis

As a transition between the former word level and the latter time level, the phrase-level framework assigns the recalculated value of POS and SO to a relevant phrase (see S2 Table.). These

values can be applied to the subsequent time level because the sequence of phrases is defined as the read time. Specific data mining approaches for the following parameters of places, story time, and *sentiment classification* scores are valuable.

**Places.** A noun-space (ns.) such as Chang'an city and specific place names inside the city such as the Buzheng Ward (布政坊) and *Xingyuan* Garden (杏园) (located in Tongshan Ward [通善坊]) account for 1.1% of the total texts tagged as the level of residential wards and streets directly mentioned (e.g., Buzheng Ward) or most likely to be located (e.g., Tongshan Ward [20]). These uniformly fine-grained places are applied to cover the corresponding story phrases of the plot that takes place in these places (Fig 3).

**Story time.** Noun-time (nt.) such as the *Tianbao period (*天宝*)*, *10 years later (*十年*)*, *more than a month later (*月余*)*, and *another day (*他日*)*, which comprises 2.7% of the total texts, is used to simulate the whole story time in an interval of every single day. The entire story timeline that the research team constructed from the texts begins from when Student Zheng enters Chang'an in 747 AD and ends around the happy ending of the novel, that is, the year Zheng is appointed to become an officer, namely, 754 AD and the year Li Wa is conferred the title *Lady Qian'guo* (汧国夫人), namely, 775 AD. The story's timeline is defined by its exact starting time during the period from 742 to 746 AD (天宝年间), its completion time by Bai Xingjian in the August of 795 AD (贞元中……乙亥岁秋八月), and other nt. phrases.

**Sentiment classification score.** The SO values (including the influence of *sentiment shifters*) of all sentiment expressions from the first to the current phrase are summarized to classify the sentiment of the already read "document", the so-called *document sentiment classification using sentiment lexicons* [18] (p.59-61). This value is the *sentiment classification* score of the current phrase.

## Time-level analysis

The purpose of time-level analysis is to unfold various structural semantics and time slice patterns computed at the space level on a read timeline. Building on the hypothesis that narrative reading is diachronic, it is worth revealing the reading mapping of variables such as the different types of POS, characters, places, story time, and the *sentiment classification* scores of the read time. At the time level, different variables of the phrase level are applied directly. A notable point about sentiment is that the curve of the *sentiment classification* score is based on the following: the accumulated SO value of the already read document will keep changing along the read timeline. This dynamic can be regarded as the integral function of SO value to the read time because time is a notable aspect in the definition of emotion [18] (p.38).

Additionally, the waves of these variables unfolding over read timelines can be compared to each other based on broad syntactic dependency to make the information much more multidimensional; for example, *sentiment classification* scores in different places become comparable.

## Space–time-level analysis

Analysis at this level unfolds various structural semantics and time slice patterns computed at the space level in the context of urban space on the read timeline. The reading of a narrative or the narrative itself is both diachronic and synchronic [12]. However, when an archaeological map (S2 File) is necessarily added to the aforementioned temporal cognitive dimension, the type of reading may go slightly beyond the scope of an individual's pattern recognition. However, visualization in this process emphasizes a multidimensional, text-accompanying, and background-based form of deep reading. This integration of multidimensionality and background information is based on a contemporary comprehension of the author's chronotope,

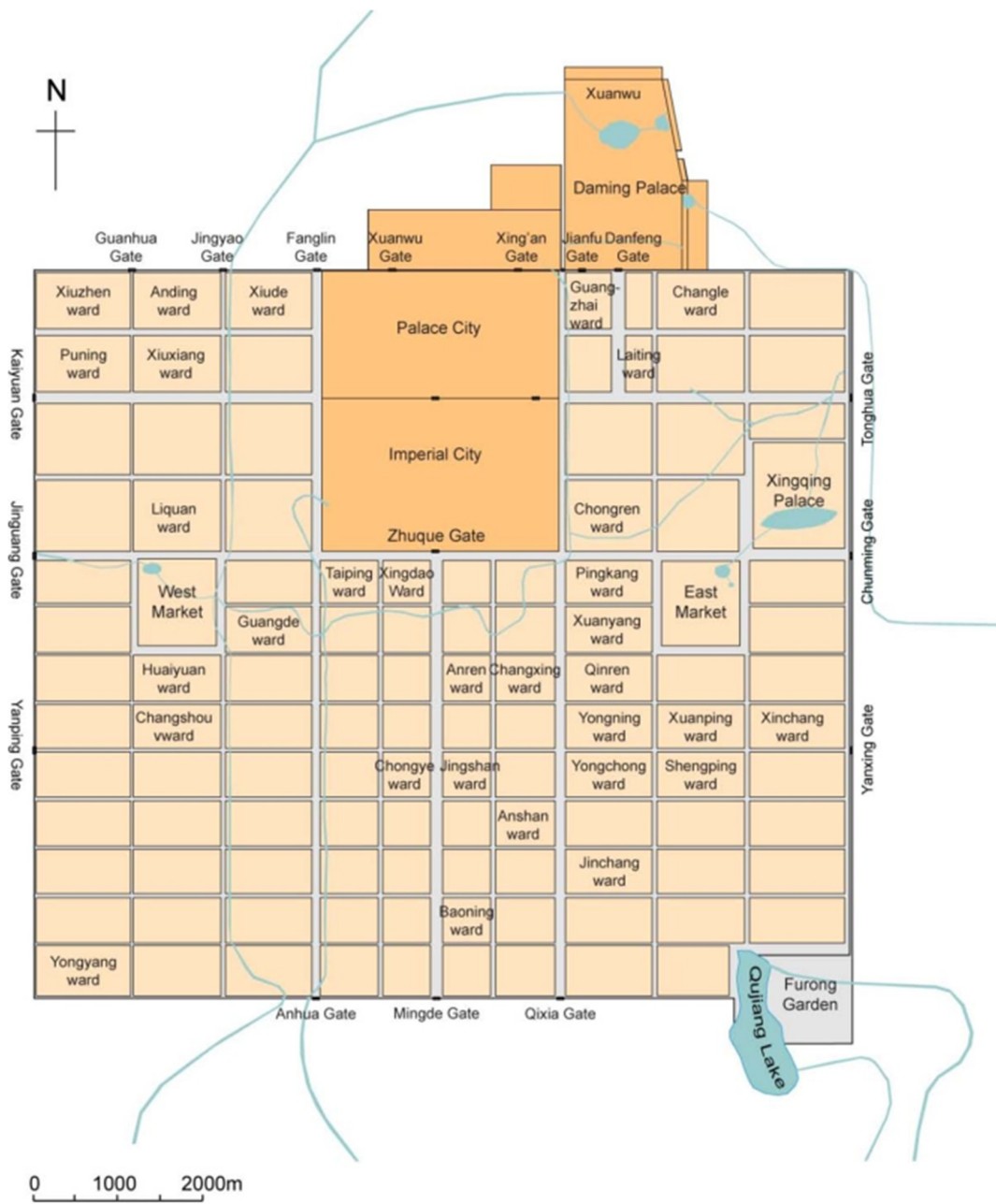

**Fig 3. Plan of Chang'an city in the Tang dynasty.** Source: [21] (p.93). Chang'an city comprises Palace City, Imperial City, and the regulated outer city composed of wards, streets, and two markets.

place anchoring of daily life (see S3 File), and routine simulation (see S4 File) within the concept of the *disciplinary space*. Fig 4 shows the integration analysis of the inherent road network of Chang'an city based on spatial syntax (see S5 File), a technology used to analyze the spatial layouts, and human activity patterns in urban areas [22]. The degree of integration (a space syntax parameter) reflects the ease of access to streets; essentially, it may determine which street is more likely to appeal to Zheng as an explorer of Chang'an.

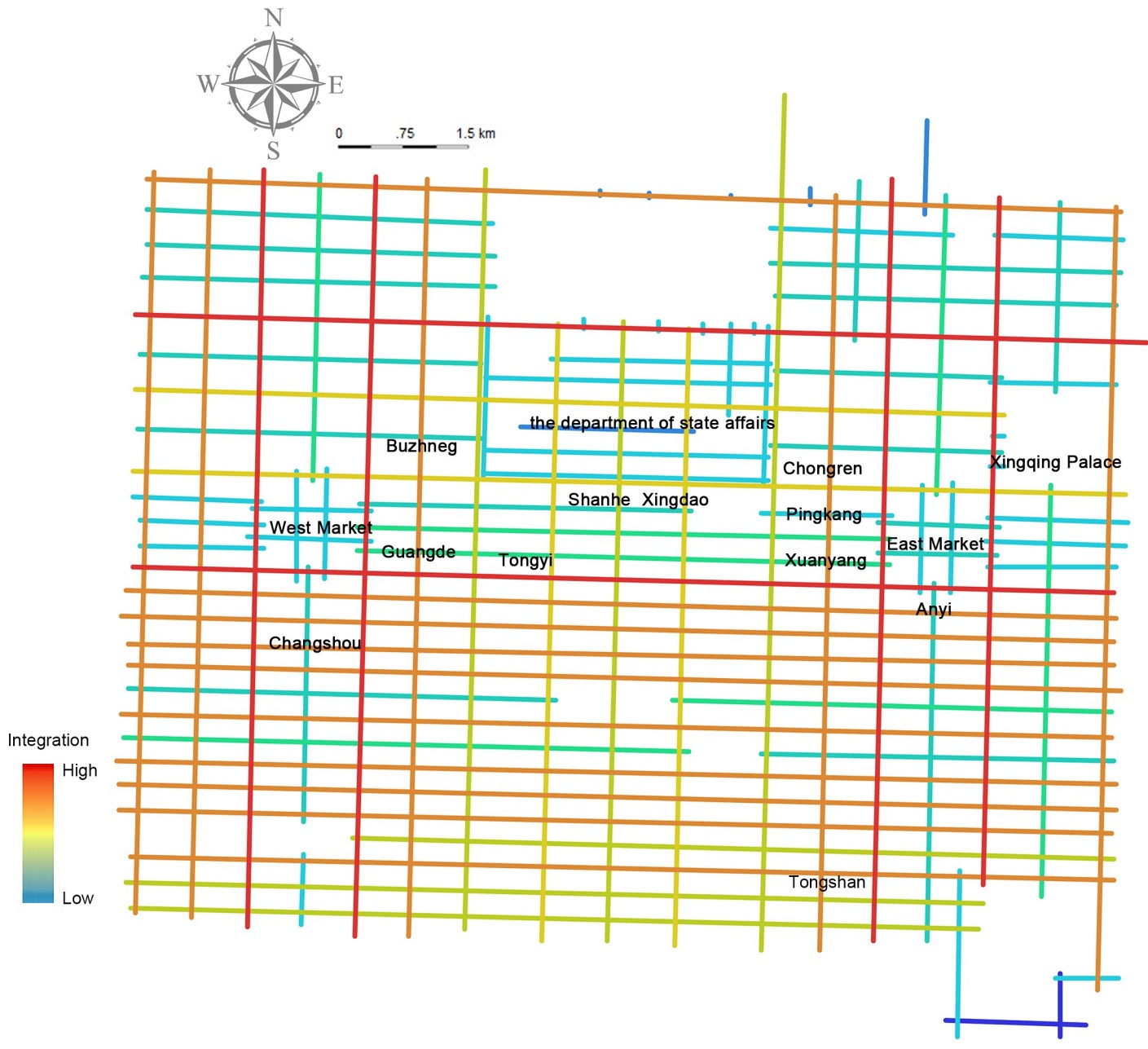

**Fig 4. Integration analysis of the road network of Chang'an city.**

## Space (map)-level analysis

This level focuses on computing the pattern of read time slices. Regarding the temporal flow of reading, in actuality, there is an evolving reading-based experience of dynamic time flow and meaningful patterns in each time slice, both of which are difficult to describe strictly. Thus, the trajectory of this pattern mapping on the read time flow can only be regarded "as part of the whole narrative" [23] (p.2). However, the narrative's psychology-based mappings are similar to the precise process control of a film's narrative by directors and actors, which conveys mixed senses to the audience.

Thus, the mappings at the space level are merely a perspective of data mining that those changing variables in the chronotope are constantly condensed into a two-dimensional and meaningful pattern through distant reading. These mappings will also be constantly unfolding in the read time to represent narratives.

## Results: Representation of narratives

In a reading process of text and digital representation of narrative, the comprehension of the story is deepened to varying degrees. This section discusses part of the large number of diagrams for the analysis and comparison, and organizes a set of subjective and experiential "what-how-why" knowledge model relating to the theme of the growth of the male protagonist that may also have other potential interpretative approaches. At the time level, the Sigmplot chart is used to show the basic variables' trajectories of read time and the point of the story. The space–time level uses ArcScene, a 3D viewer of ArcGIS [24], to unfold various variables' trajectories in the context of a two-dimensional urban space of read time as well as the story's details. A meaningful world is represented at the end of the read time at the space level, which applies statistical algorithms to the spatial analysis by ArcGIS, community detection and node centrality in network analysis by Gephi [25].

### Time: Variables' trajectory of read time

The time level shows the mapping trajectories of changes in the basic variables of the novel in the reading process, helping readers understand the narrative content, narrative theme, and the narrative skills. The representation of sentiment, characters, places, and time shows the structure of the plot, the interaction between the characters, the promotion of the plot by places, and the control of the time flow rate (details of the plot), respectively.

The value fluctuation of the integral function of the SO value is the trajectory of the *sentiment classification* score of the changing document using sentiment lexicons as read time unfolds, which indicates that the computed document at each time point is from the beginning of the story to the current read time (Fig 5). Comparing the relationship between this sentiment pattern and plot development draws its initial inspiration from the book *The Bestseller Code*, and the sentiment pattern of *The Tale of Li Wa* turns out to be an "N" pattern of a "Coming-of-Age" story that heralds a story of self-transformation [26]. *The Tale of Li Wa* is narrated from Zheng's perspective, describing his interlude in Chang'an prior to entering the bureaucracy. He experiences a happy turn of fate (falls in love with Li Wa and decides to relinquish the Imperial Examination) and then loses everything (e.g., money as well as being abandoned and beaten) before returning from despair to fulfillment (i.e., getting rid of Li Wa's madam, recovering and attaining a *jinshi* (进士, successful candidate) degree, and subsequently being assigned an official position). One reason the novel utilizes Li Wa in its title may be because the turning points of this *up-down-up* plot, as well as Zheng's strikes, are all directly led by Li Wa. The numerous slight fluctuations in the "N" curve imply small clues to the plot. The beginning of the descending section can be considered in the context of either being abandoned or the foreshadowing of the prior period—running out of money. Winning the elegy contest, the climax of the descending section, is also the beginning of the incidences where he is physically abused by his father. The ending point of the descending section can be considered in the context of Li Wa's departure from her bawd with Zheng or from the reunion of the two youths.

The combination of the instantaneous appearances of characters and value waves of the integral function of the SO value is the *sentiment classification* score of an already read document's character-attribute based on a broad syntactic dependency. Fig 6 separately inspects the

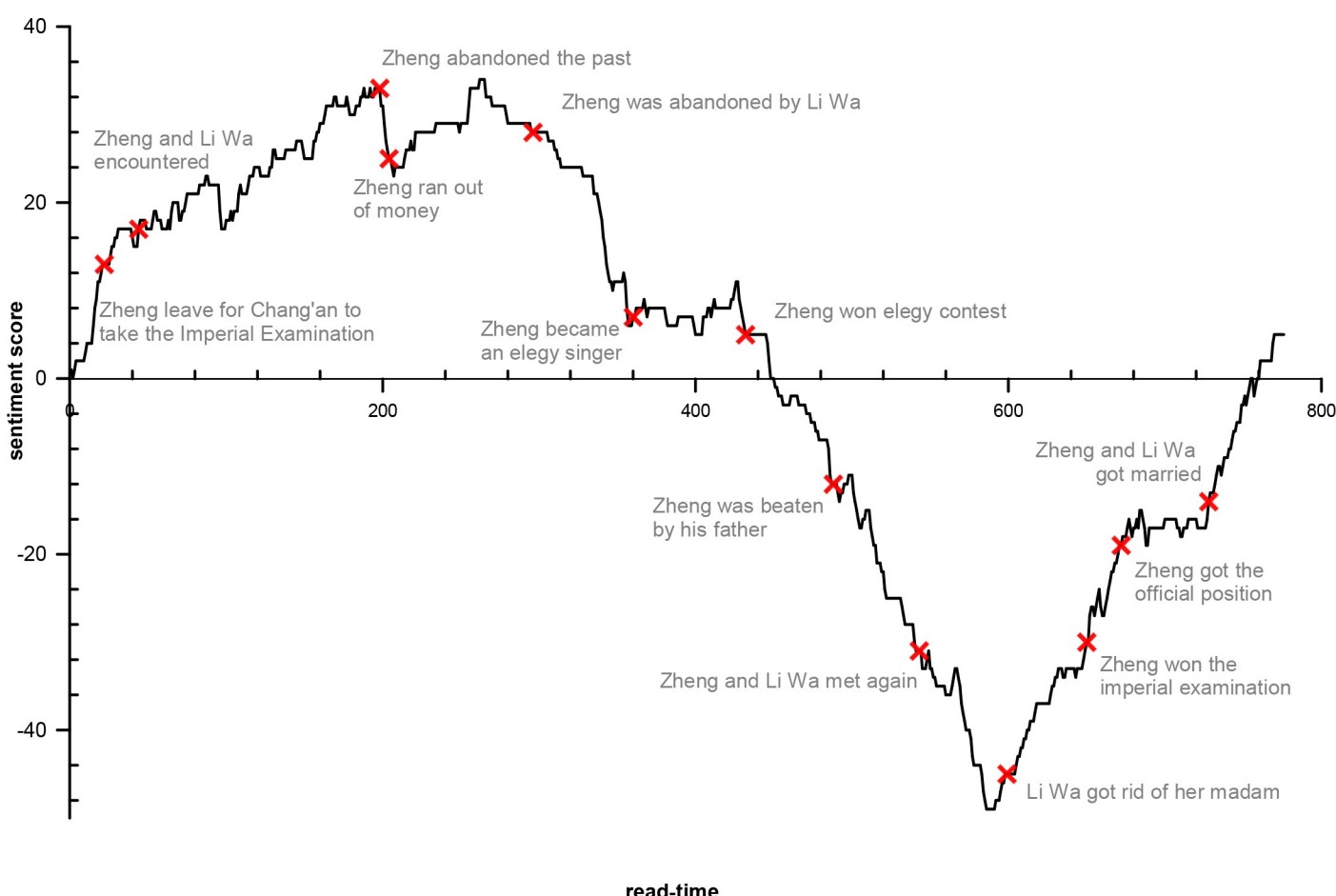

**Fig 5. Trajectory of the integral function of SO value.**

sentiment conveyed or felt when different characters appear in their read times. The emotional synergy of diverse characters partly represents the interaction between them, which typically presents the mutual checks and balances of various factors in Chang'an including love, secularity, a career, and the civilian life. The synergy of sentiment waves between Zheng and Li Wa undergoes a change of "general synergy-interruption-high synergy": when the two officially meet for the first time, they have a high degree of co-occurrence, but the sentiment synergy still belongs to "crossing swords." After living together, the co-occurrence degree is low and the sentiment synergy is relatively improved. Following Li Wa's abandonment of Zheng in the middle section, Li Wa's description and the sentiment that she carries stop abruptly, accompanied by Zheng's gradual fall to the bottom of Chang'an's society. Until the reunion of the two in the later period, when Li Wa reappears with her sudden conscience, their co-occurrence and sentiment synergy is maintained at a high level, which is a sign of the story's successful ending. Regarding Li Wa's madam, her three appearances are critical: the first two sentiment trends are the opposite of Zheng that she is successful in seducing and persuading Zheng to relinquish his original ambition. Her third sentiment trend is equivalent to Li Wa's when she is pursued by Li Wa, which leads to the two youths' successful escape from her control and the rebirth of their love. Zheng's father (荥阳公) is a local official in Changzhou (常州) and has a substantial impact on Zheng, exemplified by the three high-synergy sentiment points: before entering Chang'an, he urges Zheng to prepare for the Imperial Examination; when he meets

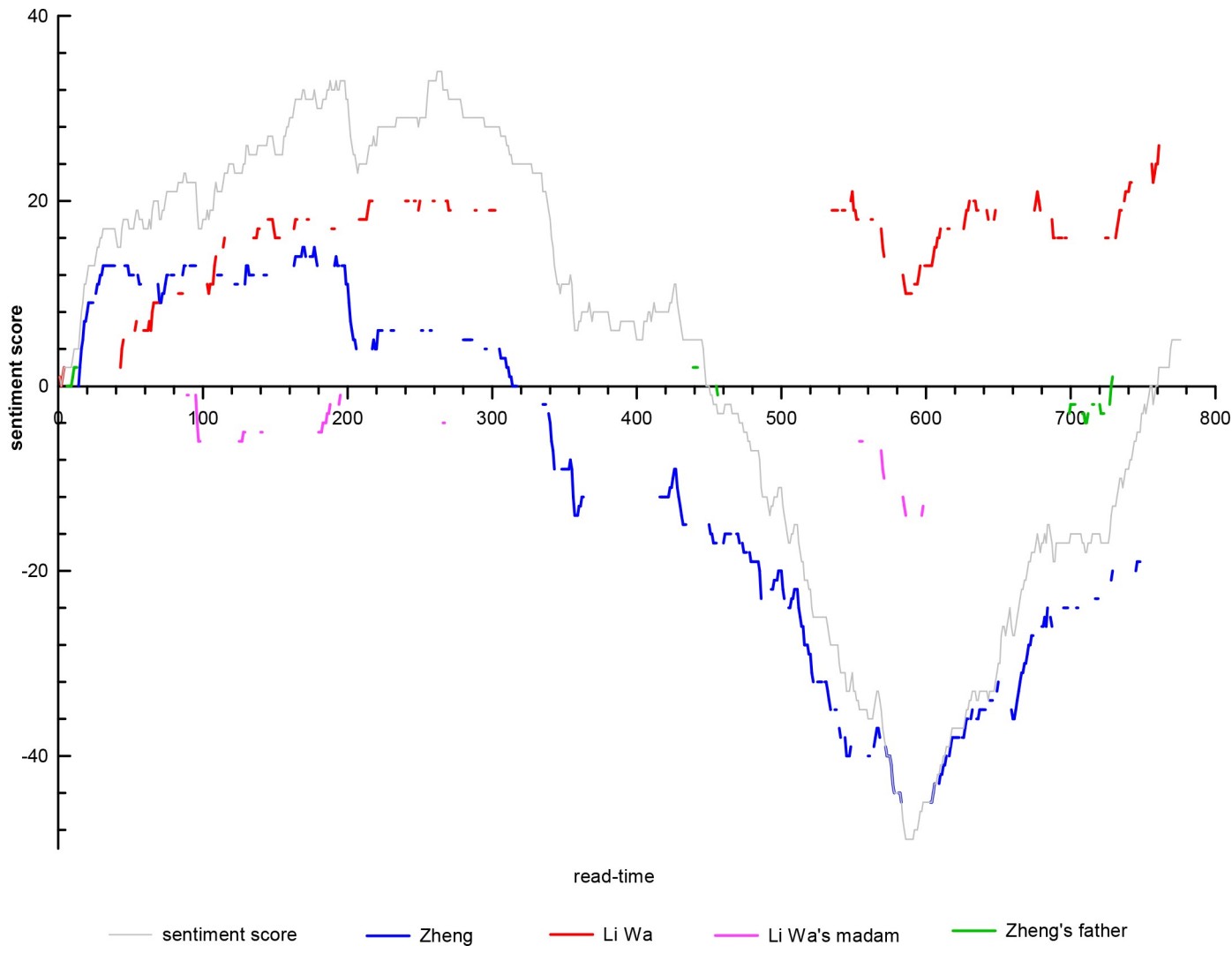

**Fig 6. Trajectory of the integral function of SO value and characters' appearance.**

his fallen son in Chang'an, he beats him nearly to death; and when he meets with Zheng in Jianmen (剑门) after Zheng attains an official position, he accepts Li Wa as a family member.

The combination of the instantaneous appearance of places and value waves of their integral function of SO value is based on a broad syntactic dependence. The story's rotation in places and conversion between places fully reflect the characteristic of its "spatial narrative" (Fig 7). Chang'an, as a bustling and complex capital city, can provide a variety of urban experience. Zheng experiences a free and complex life here apart from the Imperial Examinations, which is likewise a great opportunity for his personal growth. The main plots in the story, including the love life of the characters, the designed abandonment, and the latter reunion, are set in the following locations: Pingkang Ward (平康坊, a hooker district), Tongyi Ward (通义坊, which has a temple), Xuanyang Ward (宣阳坊, wherein a noble house for rent and an eastern government office is present), Anyi Ward (安邑坊, which has a residential house), the Department of State Affairs (尚书省, which also serves as the Imperial Examination hall), and the Xingqing Palace (兴庆宫, the royal palace where the Examinations are held). In the middle section, when Zheng gradually loses everything and is lowered to the level of the untouchable

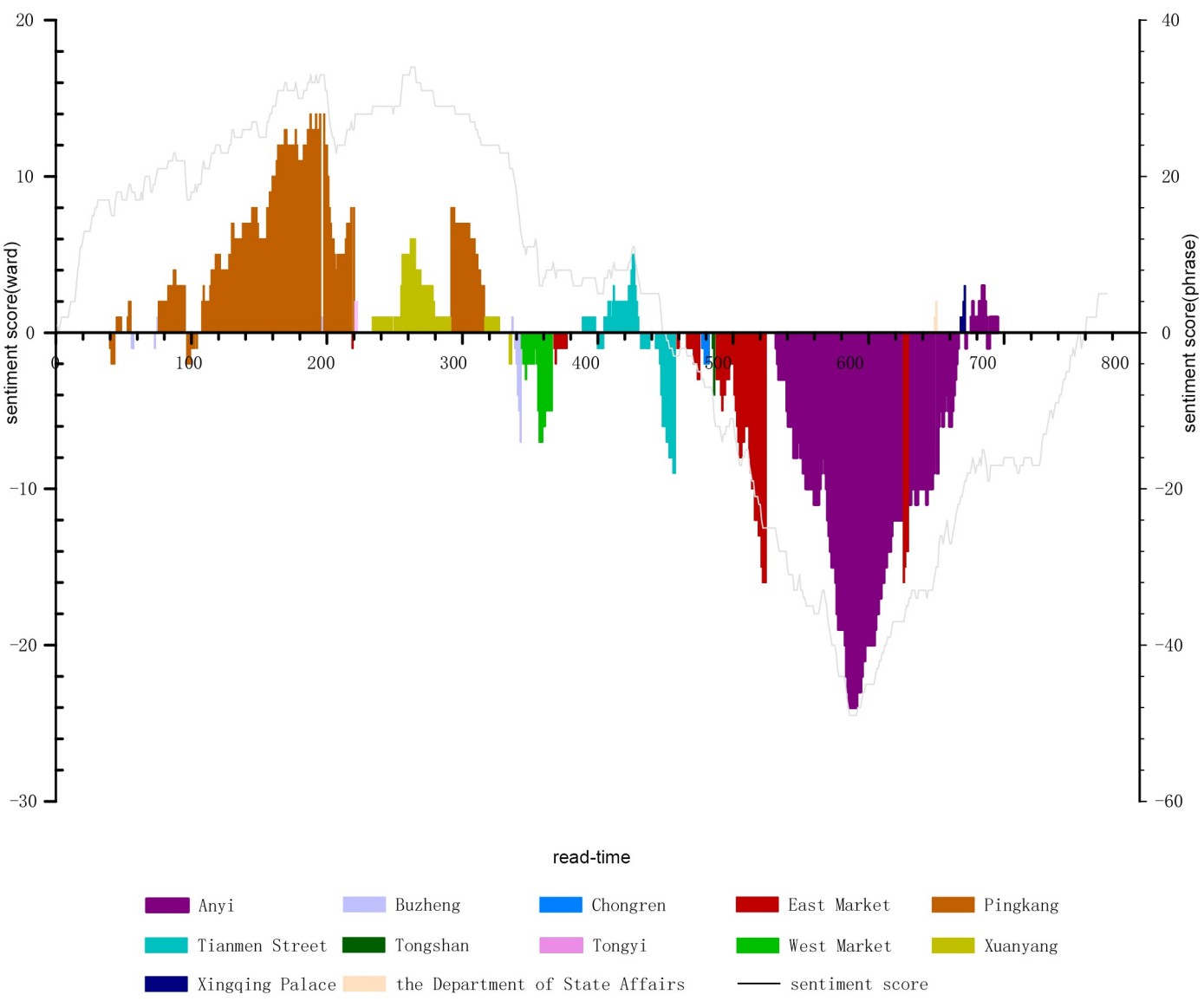

**Fig 7. Trajectory of the integral function of SO value of places.**

class in Chang'an, the plots include apastia, learning elegy, becoming a master of elegy, winning the elegy contest, being taken to his father, being beaten nearly to death, and becoming a beggar; the aforementioned events are all scattered briefly and quickly in sequential places including Buzheng Ward (布政坊, which has a house for rent), the West Market (西市), the East Market (东市), Tianmen Street (天门街, the central avenue of Chang'an), Chongren Ward (崇仁坊, which has an institution for foreign officials), Tongshan Ward (通善坊, where there is a notable scenic spot), and finally the East Market again.

The instantaneous appearance of the story time and value waves of the integral function of SO value are observed by direct comparison for the "discourse time," i.e., the rescheduled story time (Fig 8). The time continuity of the beginning and ending of the story is the consistent style of Tang Tales and indicates that the author adds his/her personal statement and judgment to the story. Initially, Li Wa's moral integrity is praised as the main writing purpose, while the source of the story and writing background are indicated at the end. The supernatural time flow

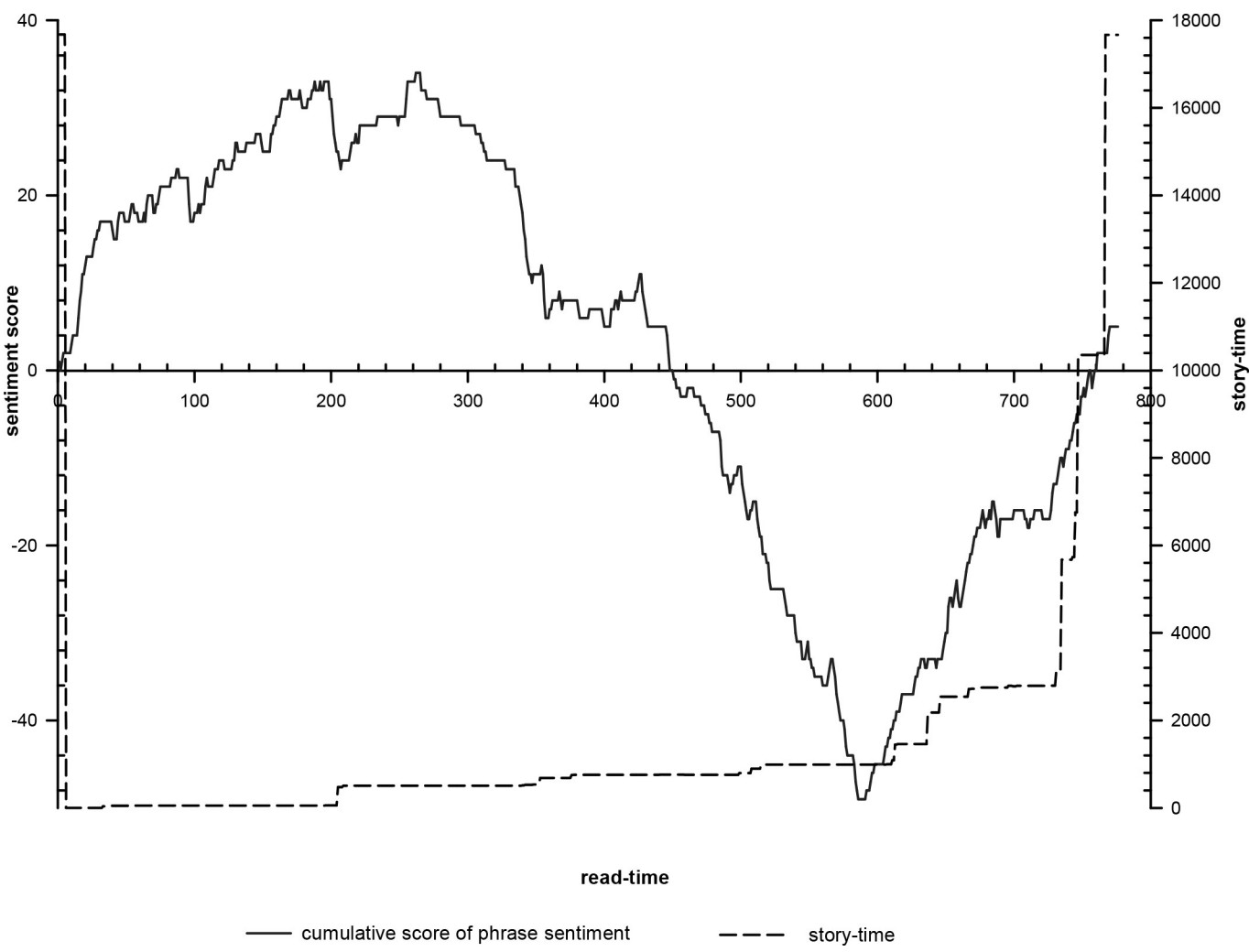

**Fig 8. Trajectory of the integral function of SO value versus the story time's appearance.**

is utilized to connect the daily story time with the story time of the beginning and ending. During the daily time, the researchers observe several fast-flowing nodes that highlight and condense the huge social role transformation of Zheng: choosing to stay with Li Wa until he runs out of money; regaining his health and learning elegy; becoming a famous elegy singer; surviving his father's beatings and becoming a beggar; being well fed by Li Wa and retaking the examination; and living his clan life after marriage. The time flow after their reunion becomes faster until it catches up with the author's final self-reporting flow rate, which in the author Bai Xingjian's eyes seems to reflect Zheng's due meaning of being a scholar, an official, and having a harmonious family after his Chang'an interlude. That is the interlude's unusual meaning.

## Space–time: Variables' trajectory of read time in the spatial context

The space–time level shows the mapping trajectories of the changes of different variables in the urban context on the read time, which helps readers gradually understand the background of the narrative of the author's experience and the inner logic from the perspective of *disciplinary space*. From urban wards and the story on wards, urban streets and the characters' mobility on the streets, urban streets and the conversion of plots, and urban streets and social class on

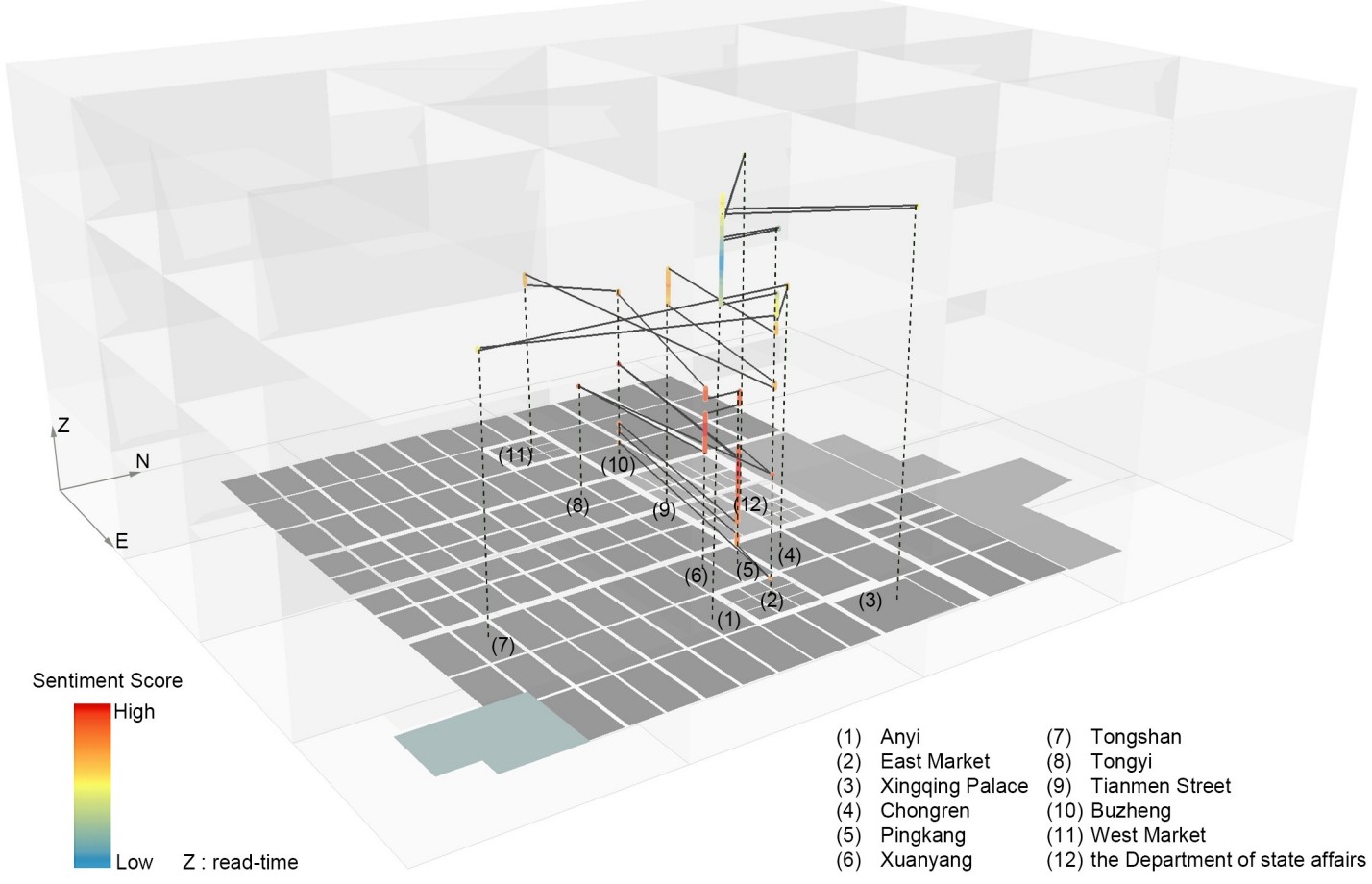

**Fig 9. Visualization of the integral function of SO value versus places' appearance.**

the streets, the author's chronotope information is progressively transmitted into the reading process to allow the reader to better understand how the story's hero lives and grows in Chang'an, a city full of rituals and civic culture.

The combination of the instantaneous appearance of places and the value waves of the integral function of the SO value in a spatio–temporal dimension (Fig 9) shows the fluctuation of the *sentiment classification* score for the changing document in different wards in Chang'an with the increasing z-axis of read time. The real Chang'an holds up to one million inhabitants and over 100 strictly controlled residential wards (坊里), each with an average area of approximately 50 $hm^2$. Such extensible two-dimensional space helps anchor the illusory stages established in the linear narrative and provides an imaginable, abstract, co-located, and logical space: a huge grid and space of walls where young tribute students such as Zheng constantly suffer from various difficulties and the storyline continues to advance.

Taking a closer look at the role of space, character path simulations based on the appearance of characters and places reconstructed by two spatial activity rules are represented on read time (Fig 10). In the reconstruction of the integration of the chronotope of the author, text, and reader, the discourse and power embodied in the physical space of Chang'an in the author's chronotope, which shapes the author's daily life and his space concept, is displayed in the reader's time through the disciplinary-space-based simulation of the characters' path in the novel. The rules of the simulation are from the characteristics of the streets: first, prefer the

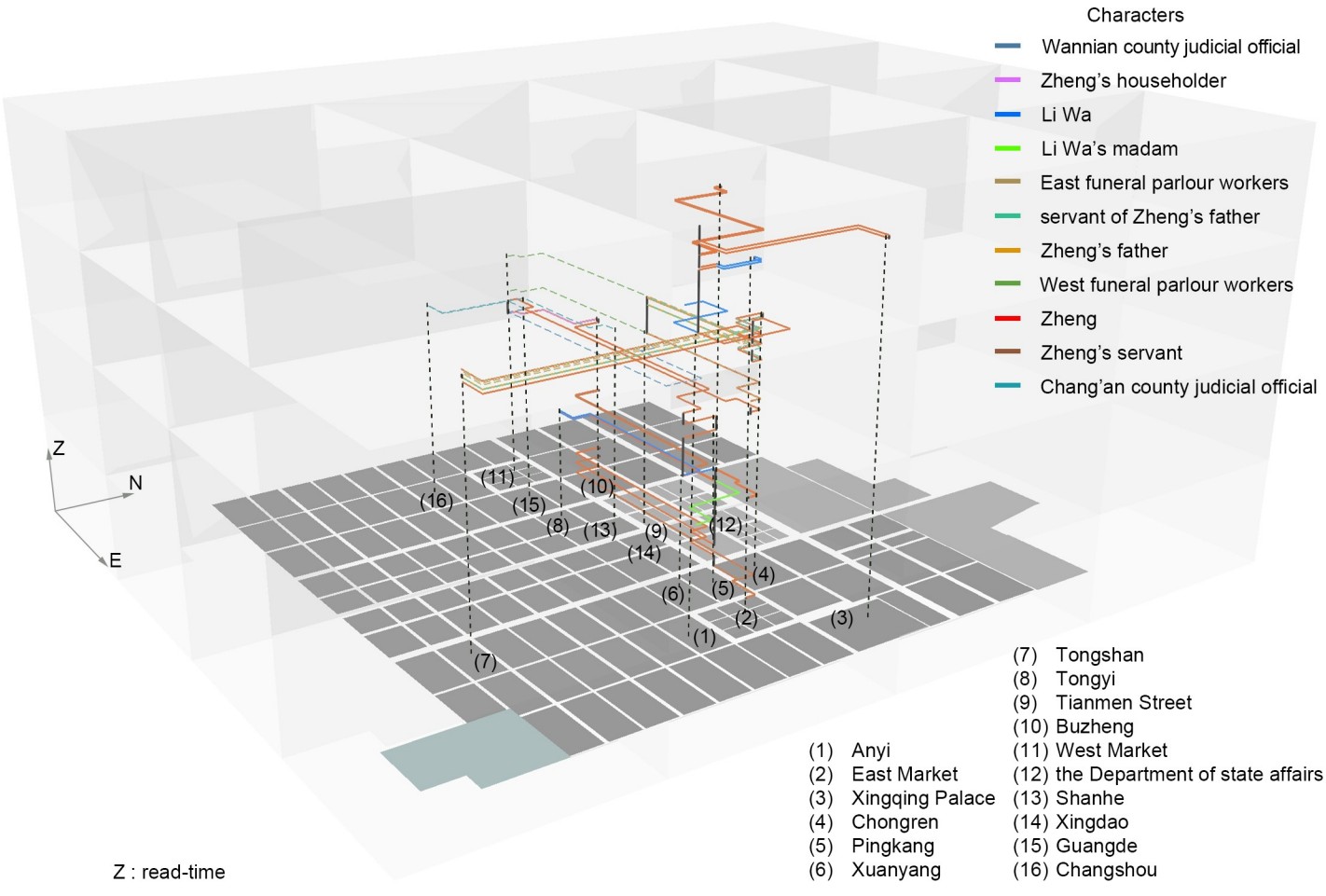

**Fig 10. Visualization of path trajectory based on *disciplinary space*: Characters' appearance versus places' appearance.**

shortest path; second, prefer the optional street section with the highest degree of spatial syntax integration. The trajectories comprising the inevitable origin-destination (OD) points and the possible paths embedded in the open street demonstrate the impetus of the geographical function separation of the metropolitan areas to the plot development and to Zheng's sharpening. For example, the isolation between Wannian County (万年县) managing the Eastern District of the Chang'an city and Chang'an County (长安县) who rules the Western District can be shown in continuing plots: when Zheng occasionally meets Li Wa in Pingkang Ward in the Eastern District and begins to miss her after returning to Buzheng Ward in the Western District; when Zheng has to abandon his hotel of Buzheng Ward in the Western District before residing in Pingkang Ward in the Eastern District; when Zheng is taken by Li Wa to burn incense and worship in Tongyi Ward in the Western District to give Li Wa's mother time to plan the abandonment conspiracy in the Eastern District; and when the funeral parlor in the East Market digs Zheng for the elegy competition, who previously serves in the funeral parlor in the West Market. Two other public places also have symbolic meaning, namely, Tianmen Street at the central junction of the inner and outer city where Zheng sings an elegy that humiliates his family and the banquet gardens for the literati (Tongshan Ward) in the south of the Eastern District where Zheng's father whips Zheng.

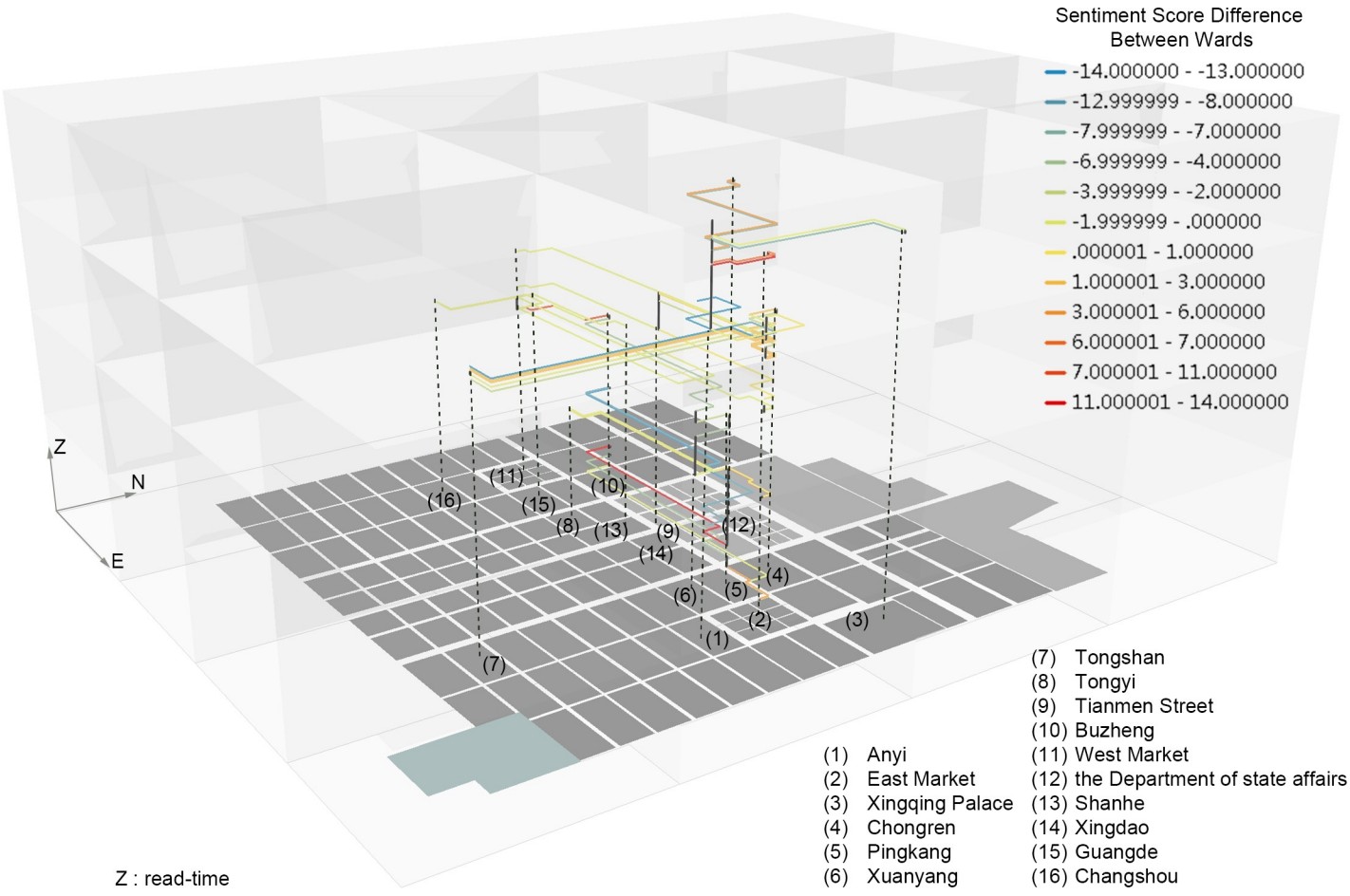

Sentiment Score Difference
Between Wards

- −14.000000 - −13.000000
- −12.999999 - −8.000000
- −7.999999 - −7.000000
- −6.999999 - −4.000000
- −3.999999 - −2.000000
- −1.999999 - .000000
- .000001 - 1.000000
- 1.000001 - 3.000000
- 3.000001 - 6.000000
- 6.000001 - 7.000000
- 7.000001 - 11.000000
- 11.000001 - 14.000000

(1) Anyi
(2) East Market
(3) Xingqing Palace
(4) Chongren
(5) Pingkang
(6) Xuanyang
(7) Tongshan
(8) Tongyi
(9) Tianmen Street
(10) Buzheng
(11) West Market
(12) the Department of state affairs
(13) Shanhe
(14) Xingdao
(15) Guangde
(16) Changshou

**Fig 11. Visualization of path trajectory based on *disciplinary space*: Characters' appearance and the integral function of SO value versus places' appearance.**

In Fig 11, the path visualization added by the places' integral function of SO value is to give a prediction of the plot change or conflict between places (blue and red represent changes) from the perspective of the path between places. The difference of *sentiment classification* score between the former place and the latter is assigned to the character's path linking them. The blue and red paths demonstrate seven dramatic plot changes with places: a positive love life from Buzheng Ward to Pingkang Ward, with Zheng seeing his love; a negative brothel life from Pingkang Ward to Buzheng Ward, with Zheng packing his past luggage to abandon his career; a negative abandonment conspiracy from Pingkang and Xuanyang Wards to Anyi Ward, with the leaving of Li Wa and her bawd; a positive civilian life from Buzheng Ward to East Market, with Zheng learning the elegy after the hunger strike; a negative beggar life from Tongshan Ward to the East Market, with Zheng rescued and abandoned again after a deterioration of his health; and a positive tribute student life from Anyi Ward to the East Market, with Zheng's rehabilitation and reactivation of his preparations for the examination.

In Fig 12, the visualization of the path trajectory added by characters' social strata and places' integral function of SO value shows the simple society in the story, revolving around Zheng whose social stratum keeps changing and vividly reflects the complexity of citizens' lives. The classification of social strata in the story from untouchable to noble is as follows: level 1, beggar, servant, or sex worker; level 2, businessman, civilian, or bawd; level 3, ward

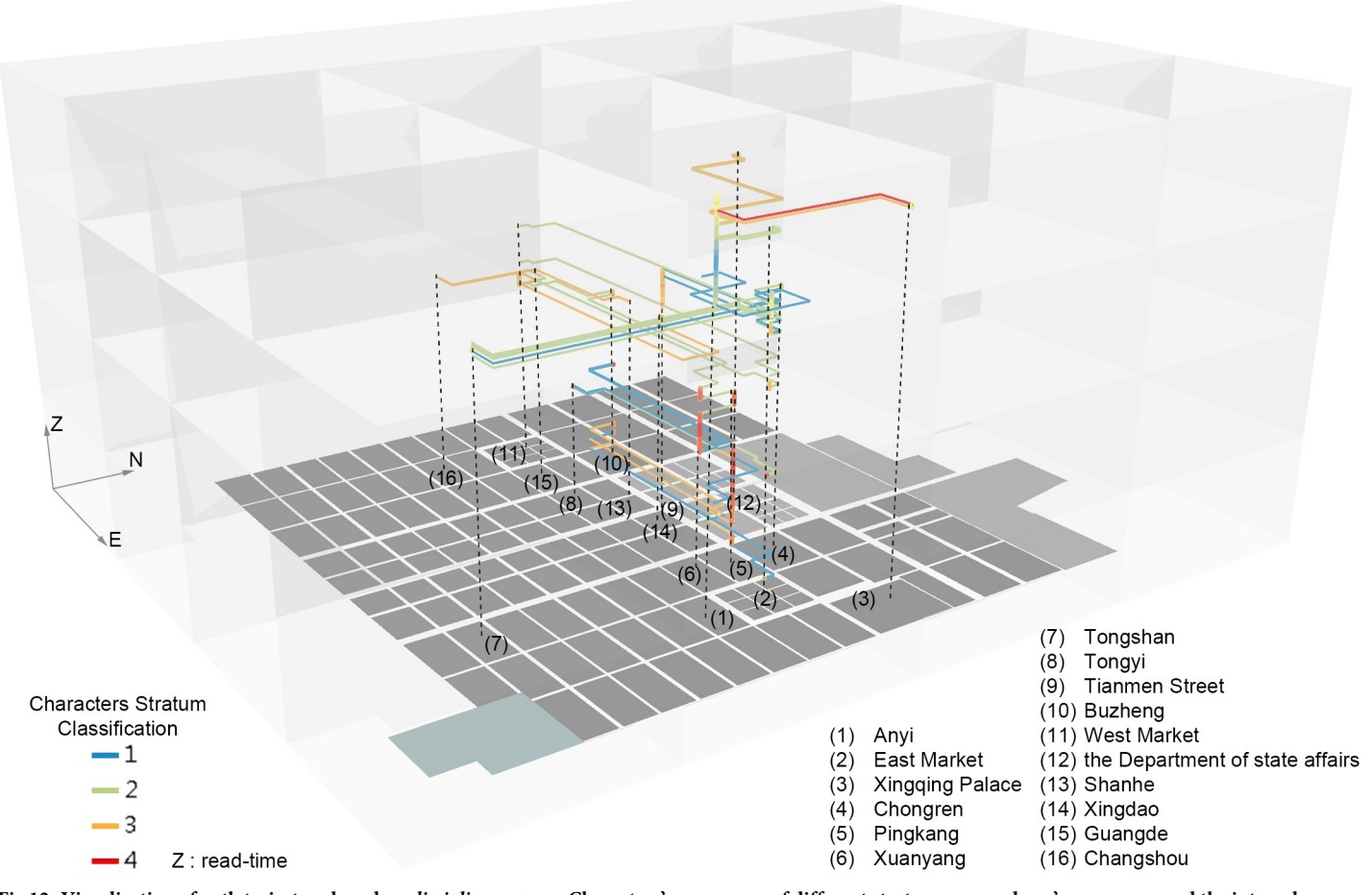

**Fig 12. Visualization of path trajectory based on *disciplinary space*: Characters' appearance of different stratum versus places' appearance and the integral function of SO value.**

head (里胥), candidate student, successful candidate, or county judicial official (贼曹); and level 4, Chang'an officials (京尹) or officials from other places. The strong correspondence between most of the social stratum values and the *sentiment classification* scores in the spatio–temporal dimension reflects the ever-changing space as an intermediary between changeable social strata and plots. And Zheng's growth is closely associated with the different social roles he experiences and plays in various places. Notably, the time when Zheng and Li Wa share an early, short life together and its high sentiment values lagging behind lower social strata can be considered as a special illusory stage of escape and the flashpoint of the subsequent stratum changes.

## Space: Readtime–compressed pattern of meaning

The space level shows the statistical pattern of mapping trajectories of changes of different variables in the urban context after the complete reading, which helps readers understand the author's experience of Chang'an and the story's setting of the background wherein a future bureaucrat is shaped. Through numerous statistics about POS, sentiment, characters, and places, the reader will grasp Chang'an's scenery and customs, the complex settings of plots and places, and the relationship among characters, social backgrounds, and spatially-embedded social relationships.

Statistics of different places' POS reflect the picture scroll of Chang'an city; namely, the function, environment, and characters' living conditions of different wards in the city. Pingkang and Anyi Wards are notable locations for the scenes of life: the former is romantic, comfortable, and well-arranged with open doors, a meeting house, meeting rooms, a curtain-couch hall, and yards; the latter is a closed residential area with exquisite food (Figs 13 and 14). This change in residence reflects the psychological transformation of Li Wa, the residences' decorator, and corresponds to Zheng's changes in mental state from the former energetic and strong emotion to the latter rehabilitation after a serious illness (Fig 15). To accentuate the abandonment conspiracy, it also maximizes Xuanyang Ward's superior, tasteful living environment along with its high-grade luxury represented by oversized doors, partial courtyards, pavilions, pools, and bamboo forests wherein Zheng then became confused (Figs 13, 14 and 15). The above three different styles of residential areas were authentic backdrops for the dramatic scenes. The East Market is an indispensable functional location and has the storefront, bookstore, management area, market entrance, and basement wherein Zheng becomes a beggar at the lowest point in his life (Figs 13 and 15). The descriptions of the areas outside the Eastern District are rather thin. The only cases include Zheng's pride of self-reliance at the funeral parlor in the West Market (Figs 13 and 15), Zheng's resentment of self-destruction at the rental house in the Buzheng Ward (Figs 13 and 15), and a singing game loser's shame on Tianmen Street (Fig 15).

The spatial analysis based on places' *sentiment classification* score after extracting the read time reflects an embedded sentiment trend in Chang'an (see S6 File). This statistic uses interpolation analysis to create a continuous (or predicted) surface from sampled point values. The geographical distribution of *sentiment classification* scores presents a complex and staggered pattern (Fig 16): The triangles in the northwest, southeast, and east of Chang'an have negative values, and the east–west triangles near Imperial city present positive values that are closely related to the story's inherent logical arrangement. This *sentiment classification* score is essentially consistent with the statistics of characters' situation in Fig 15: Negative sentiment expression comprises Buzheng Ward, the East Market, and Xuanyang Ward while the West Market is positive; Pingkang Ward and Anyi Ward are positive overall based on their composite emotions. The Department of State Affairs, Xingqing Palace, and Tongyi Ward (none shown in Fig 15) indicate a positive sentiment behind the respective plots of passing the Imperial Examination, passing the Palace Examination, and praying for getting a son. Chongren Ward and Tongshan Ward, which are far apart in distance, are also not set forth in Fig 15. Their negative plots refer to Zheng's being scolded and beaten by his father, respectively.

Character statistics enable a network analysis of the characters' co-occurrence network in the entire document in Figs 17 and 18 (see S7 File), which indicates a network around hubs (a few nodes have much more connections/edges). By measuring figures with their value of betweenness centrality [28], the researchers observe that Zheng (196) and Li Wa (63.5) are the main and secondary characters, respectively. Li Wa's madam, Li Wa's "aunt", the chief of the east funeral parlor, and Bai Xingjian (21) are also key figures that advance the plot. In the ego network (i.e., all connections with this node) with Zheng as the core, different communities can be observed: Li Wa's group, the funeral director's group, and Zheng's group. However, the top-down model (Fig 17) indicates a more harmonious network in that Zheng's father and Zheng are from the same class while the bottom-up model (Fig 18) implies Zheng's embarrassing situation between his father and Li Wa, which is somewhat similar to the different views put forth by scholars with different perspectives.

According to preliminary statistics on the social stratum values (defined in Fig 12) of the characters in different locations, changes in the social background around Zheng can be visualized (Table 1). The average values of social stratum in most of the wards are approximately 2

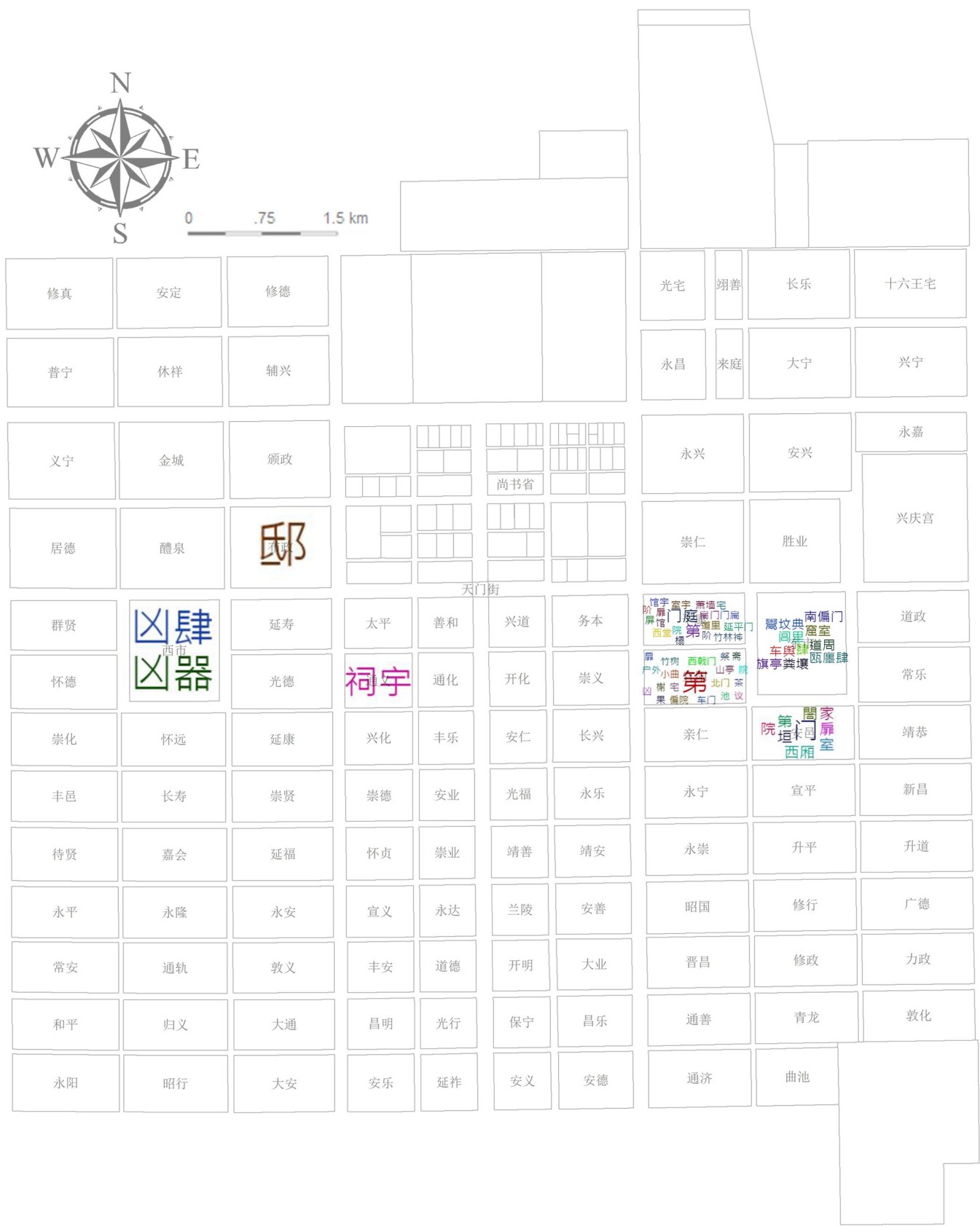

**Fig 13. Statistics of the POS and locations (noun-space in the whole document).**

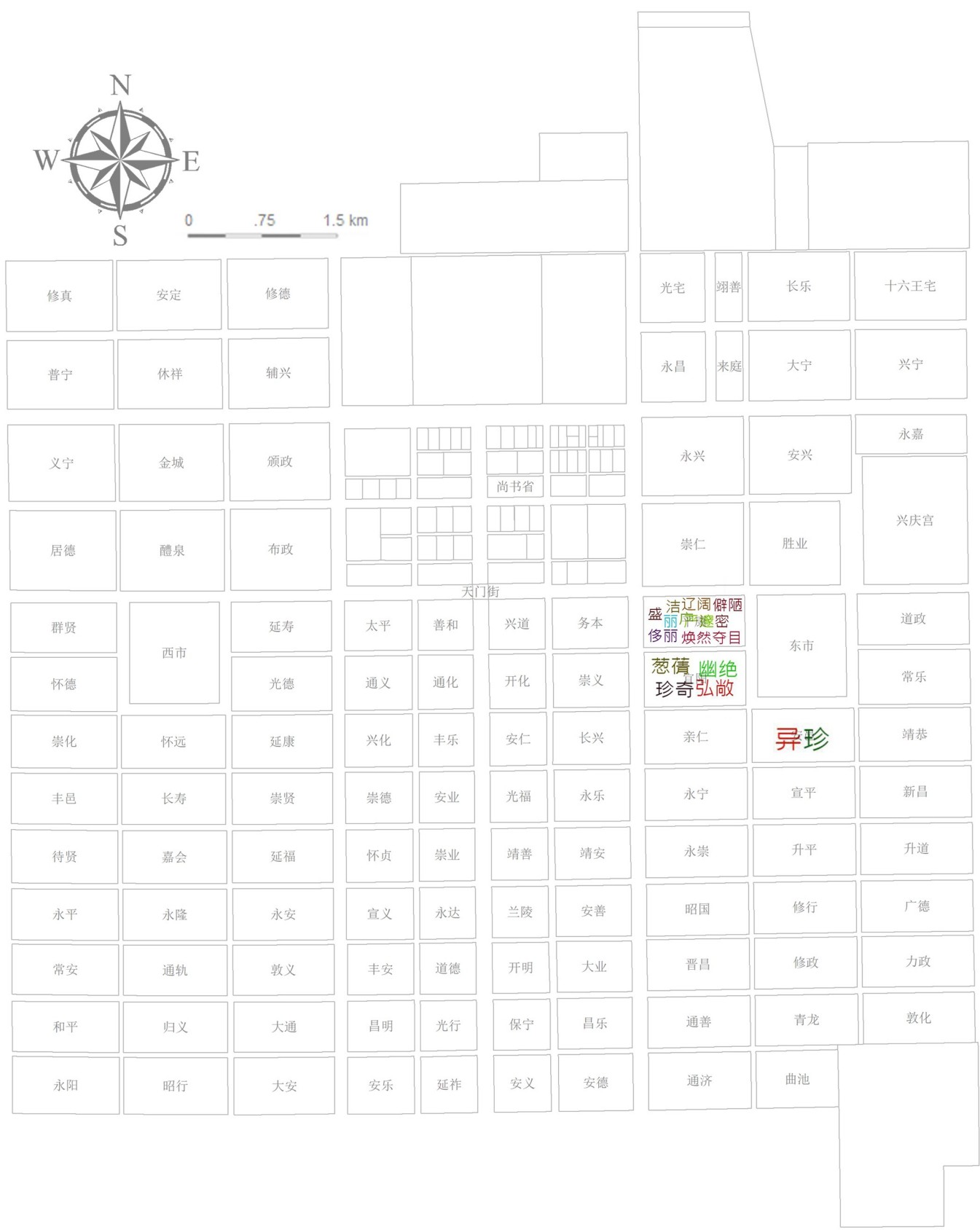

**Fig 14. Statistics of the POS and locations (adjective-space in the whole document).**

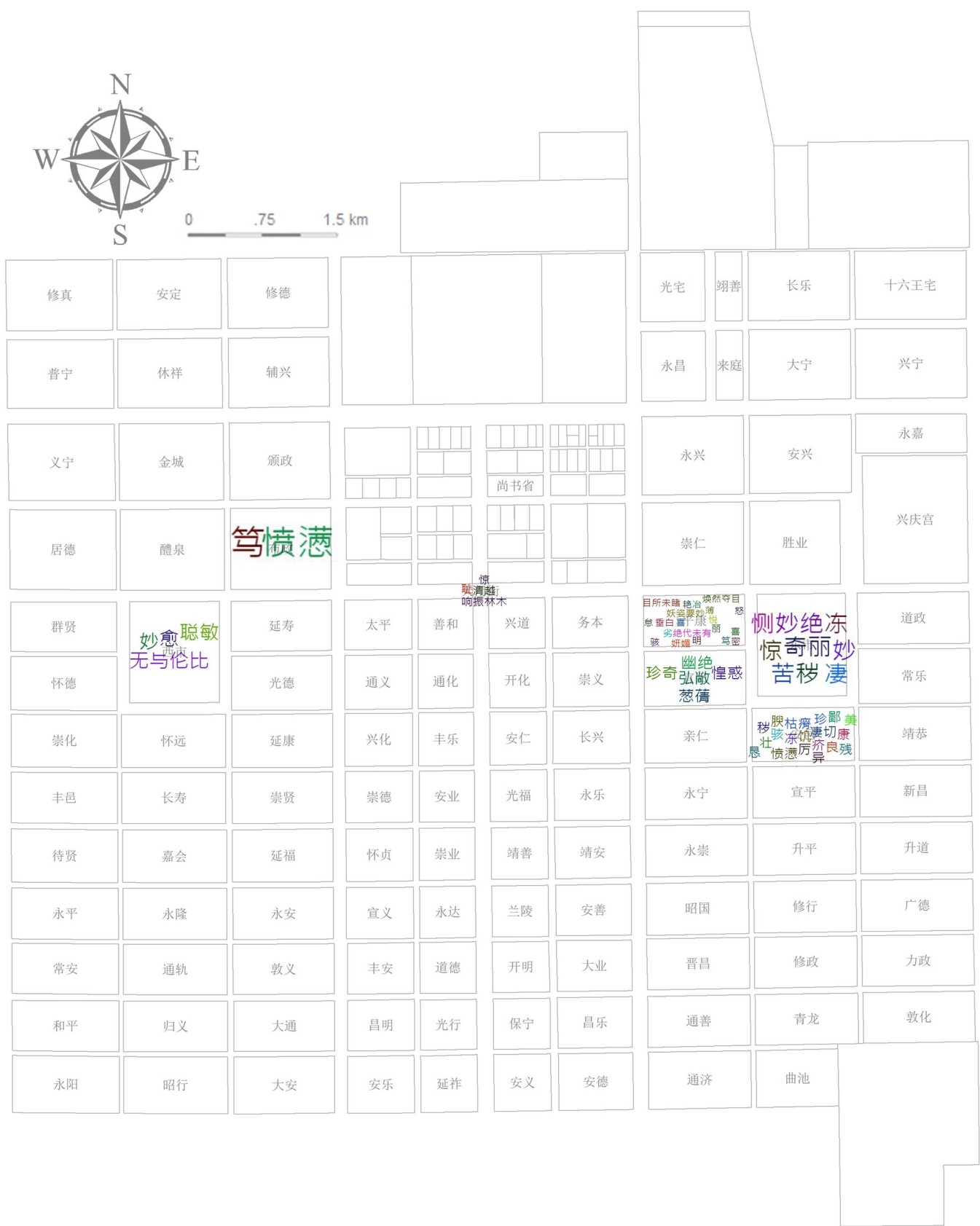

**Fig 15. Statistics of the POS and locations (adjective-character in the whole document).**

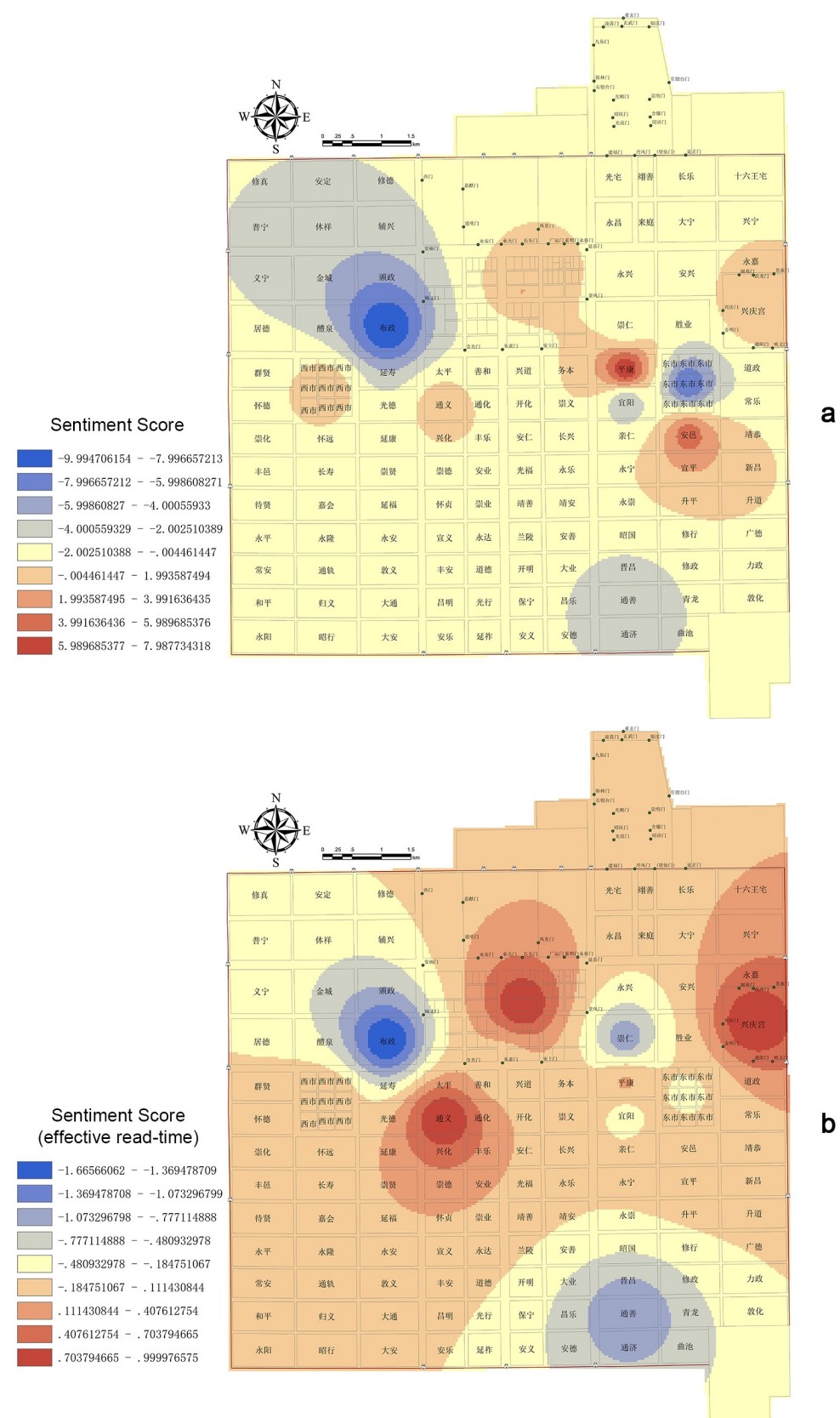

**Fig 16. Inverse distance weighted interpolation by ArcGIS [27]** of the *sentiment classification* score in places (a. *sentiment classification* score of place attribute, b. *sentiment classification* score/effective read time of place attribute).

with large variances, which is a relatively mixed background setting. The places with less appearance and less variance (Count < = 5 and StdevP < = 0.5) serve as an abstract background outside the main story segment and have a more inclined stratum value, e.g., Xingqing Palace is 3.5 (successful candidate, official); Guangde Ward, Changshou Ward, Shanghe Ward, and Xingdao Ward are 3 (ward head and county judicial official); Department of State Affairs is 2.5 (civilians, successful candidate); and Tongyi Ward is 1.5 (civilians, prostitute).

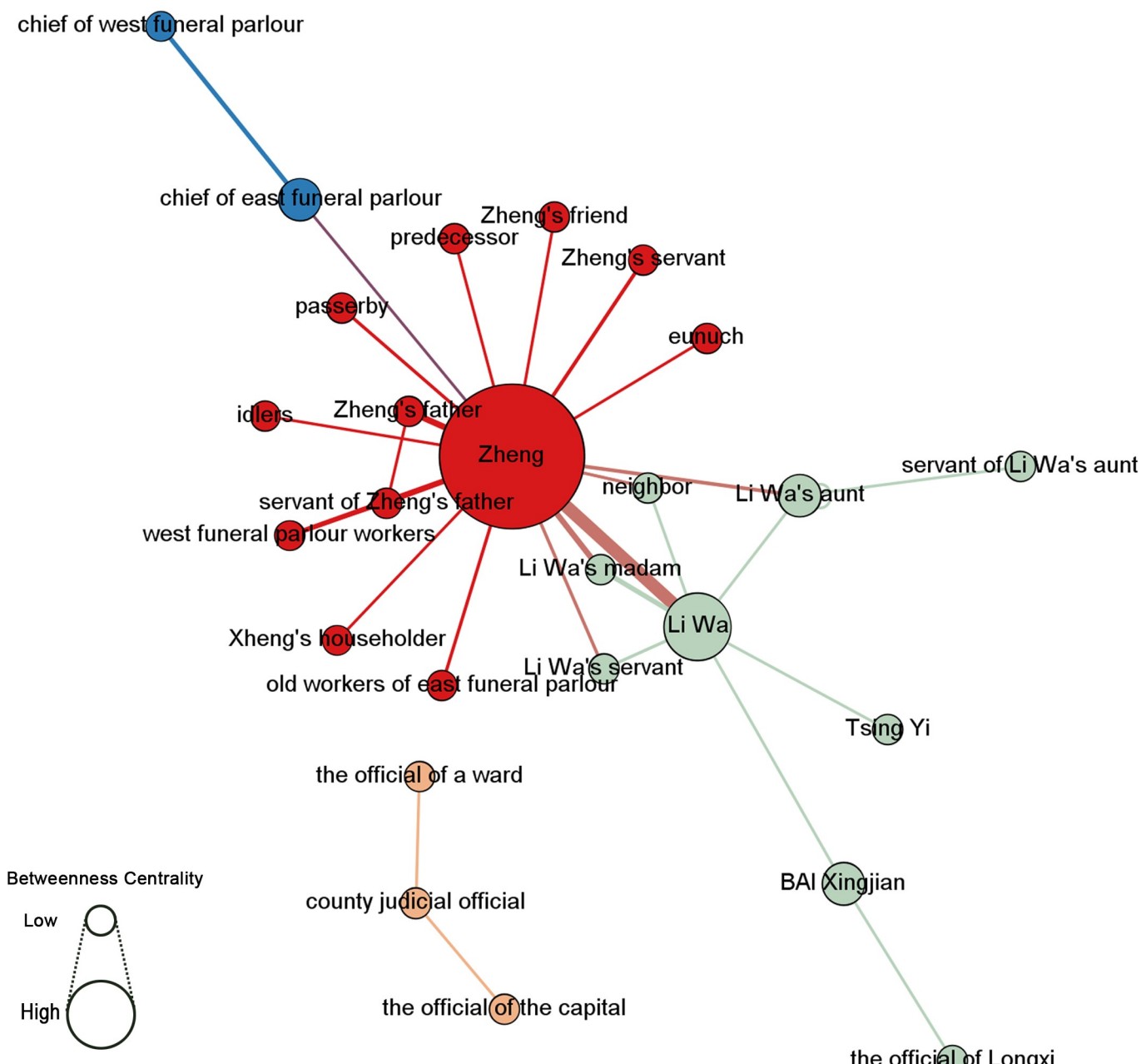

**Fig 17. Statistics of characters in co-occurrence network, Girvan–Newman clustering [29], and betweenness centrality.**

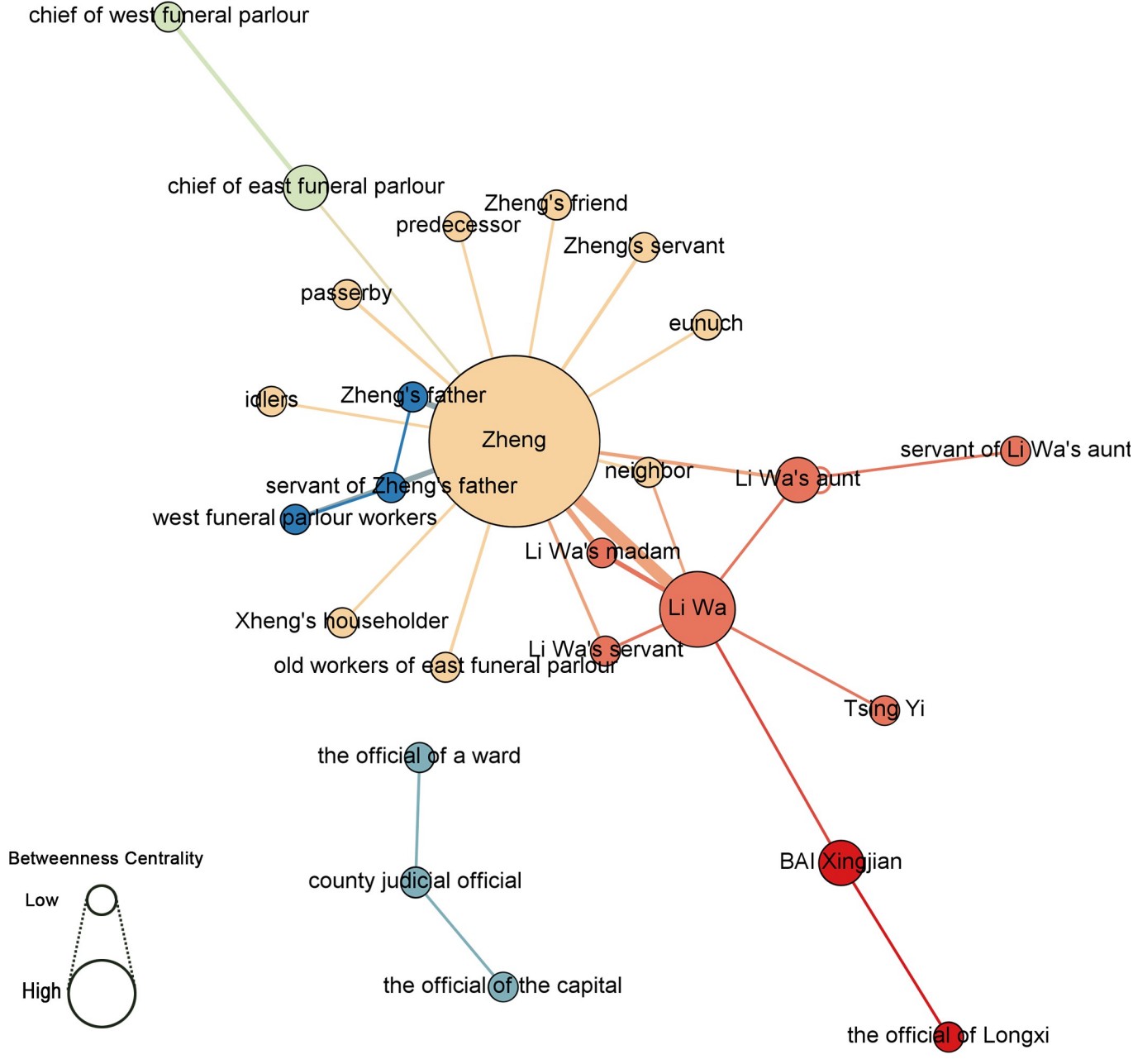

**Fig 18. Statistics of characters in co-occurrence network, modularity class [30], and betweenness centrality.**

A spatially-embedded network of characters in the entire document (Figs 19–21) is analyzed through the two-model network (places as nodes and the flow of characters between places as edges) and demonstrates that the social relationship projected into space in the story is the small world network (small average path length of 2.265625, larger clustering coefficient on 0.474)—the stable and fast-spreading metaphorical structural relationship between the places in Chang'an city (see S8 File). The ranking of the weighted degree of nodes representing the number of connections with other nodes is East Market >> Xuanyang > Buzheng = Anyi >. . .. The discourse in the East Market is comparatively less, but it becomes the most active node, and a hidden "celebrity" existing in the texts. The modularity class [30] applied to

**Table 1. Statistics of characters and locations (social stratum of characters in each location, weighted by the number of times characters enter places).**

| Place | Stratum Value- Average | Stratum Value- Variance | Stratum Value- Count |
|---|---|---|---|
| Xingqing Palace | 3.5 | 0.5 | 2 |
| The Department of State Affairs | 2.5 | 0.5 | 2 |
| East Market | 1.88 | 0.53 | 24 |
| Chongren | 2.08 | 1.19 | 12 |
| Xuanyang | 2 | 0.71 | 8 |
| Pingkang | 1.88 | 0.83 | 17 |
| Anyi | 2.09 | 0.9 | 11 |
| Tongshan | 2.13 | 0.78 | 8 |
| Buzheng | 2 | 0.85 | 14 |
| Tianmen Street | 2.2 | 0.98 | 10 |
| West market | 2 | 0 | 6 |
| Tongyi | 1.5 | 0.5 | 4 |
| Guangde | 3 | 0 | 2 |
| Changshou | 3 | 0 | 2 |
| Shanhe | 3 | 0 | 1 |
| Xingdao | 3 | 0 | 1 |

community detection divides Chang'an into four groups: Anyi, Xinqing Palace, and the Department of State Affairs, which are red power communities (class = 3); local officials of orange communities (class = 2) are concentrated in wards of Xuanyang, Tongyi, Guangde (光 德坊, wherein the Chang'an government office is located), Xingdao (兴道坊), Changshou, and Shanhe (善和坊); green-colored communities (West Market, Pingkang, and Buzheng, class = 1) represent places where individuals from other places hang around; and blue-colored communities (class = 0) represent the concentration of officials from other places to Chang'an in the East Market, Chongren, Tongshan, and on Tianmen Street.

The betweenness centrality [28] value of Xuanyang is higher than that of other places where Zheng is designed to be abandoned. This value shows Xuanyang as a "spread bottleneck," which indicates Xuanyang's control of individuals' movements. This phenomenon may be partly attributed to the short-term rental function of the ward's mansion—the smoke screen can be formed here and the story goes to the subsequent stage. Xuanyang–Anyi–Pingkang–East Market has the highest closeness centrality [28], which represents their best perspective on personal mobility and is the easiest place to obtain information, which is highly consistent with the traditional information network among brothels, officials, and civilians. By using a hyperlink-induced topic search algorithm [31] to the direct network, the researchers observe that the authority is the highest in East Market–Xingqing Palace–the Department of State Affairs and the hub is the highest in Anyi, which appears to be linked with the value orientation in the author's chronotope that Zheng returned to the right path after this interlude.

## Conclusions

This paper reconstructs and attempts to represent the chronotope information of the text, author, and reader through digital technology and visualization to improve the comprehension of the narrative of *The Tale of Li Wa*. This semi-theme-oriented representation fits Moretti's description on modern European cases that "really often felt like so many experiments. . . variables that I kept changing and changing. . ." [14] (p.4). Through the continuous chart comparison in the "Time–Space–time–Space" circle, the research team have mastered the narrative, the physical background of Chang'an city, the life of Chang'an's residents, and thus move from

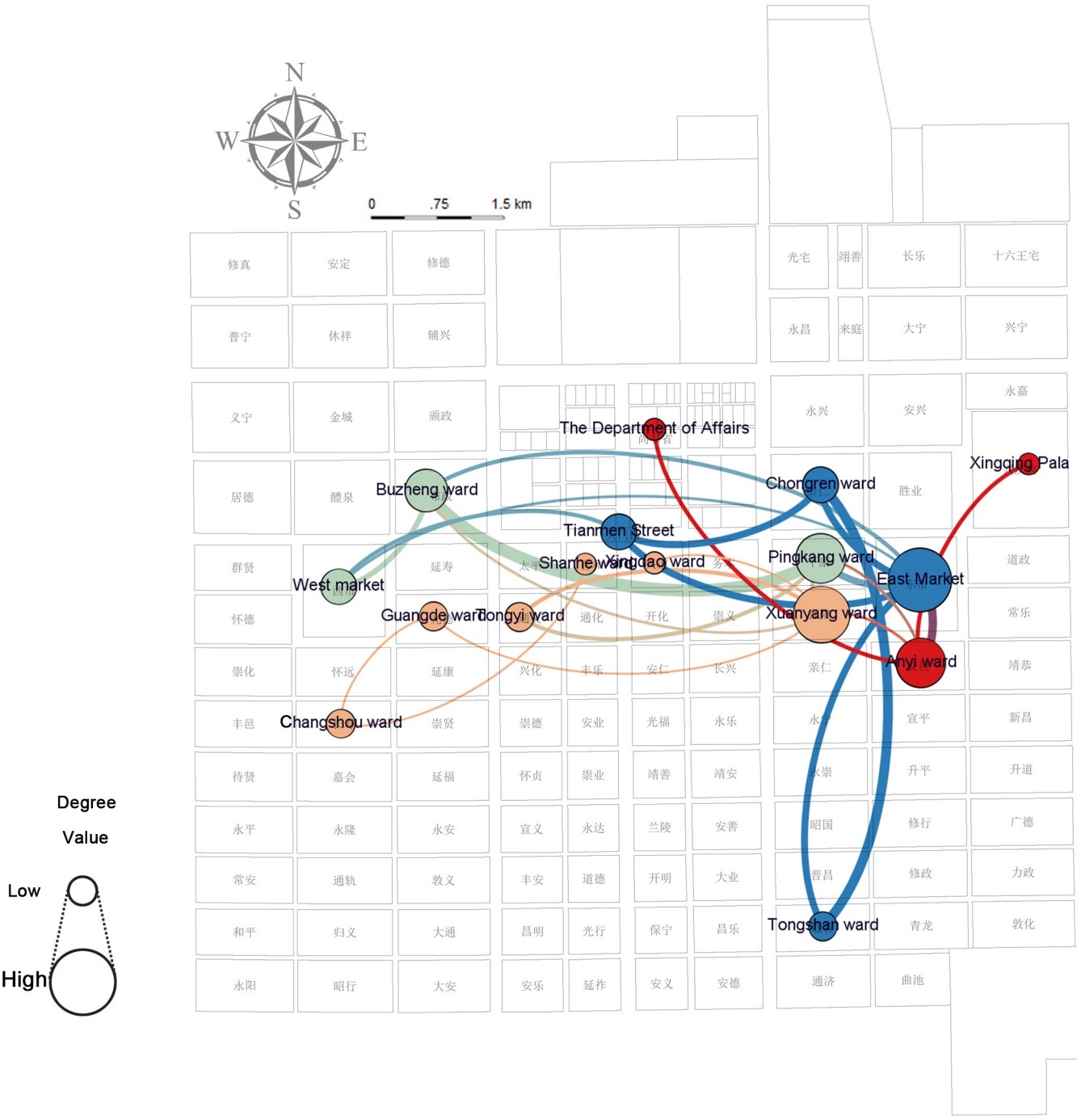

**Fig 19. Network analysis of characters and places in modularity class analysis (use edge weights: On) and weighted degree centrality based on a full-text, spatially embedded, undirected network of characters.**

a shallow to a deep understanding of one of the interesting themes, from the abstract to the concrete—the inevitable growth of the protagonist who first enters the capital Chang'an. The representation of the whole narrative once again proves the typicality and particularity of *The Tale of Li Wa* in literature and history (even if compared to traditional analytical methods, many of the discoveries herein are new): the narrative is undoubtedly strong in terms of

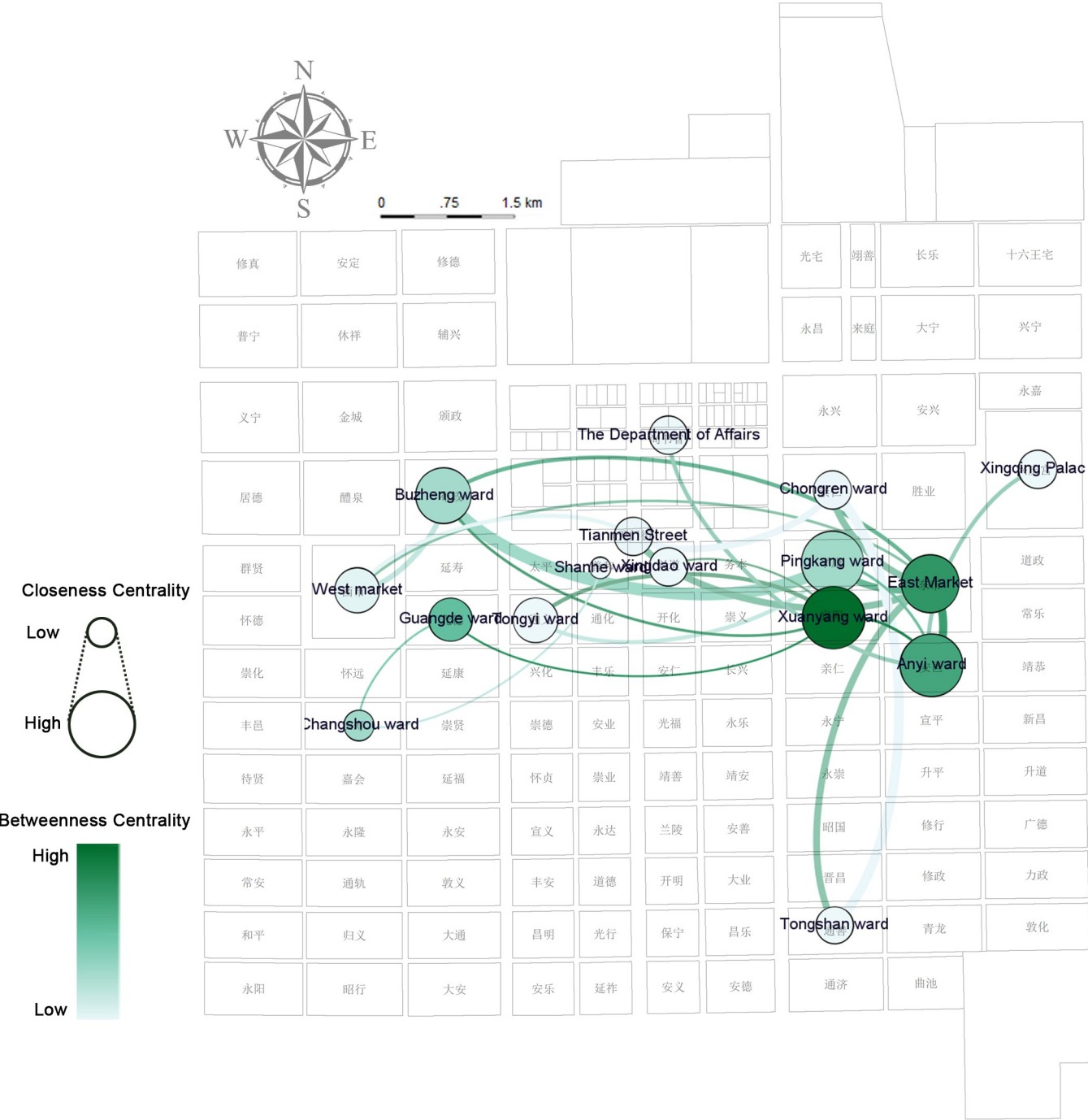

**Fig 20. Network analysis of characters and places in closeness centrality and betweenness centrality based on a full-text, spatially embedded, undirected network of characters.**

characters, plots, time, and space; the primitive image or some metaphor embodies a social culture that generally belongs to the literati in the Tang dynasty; and individuals' multiple lives in the capital city and the power controlling behind are depicted. After tracing the author's experience, it is observed that Bai Xingjian is indeed a typical official who passed the Imperial Examination, entered the bureaucracy in the middle of the Tang dynasty, and lived in

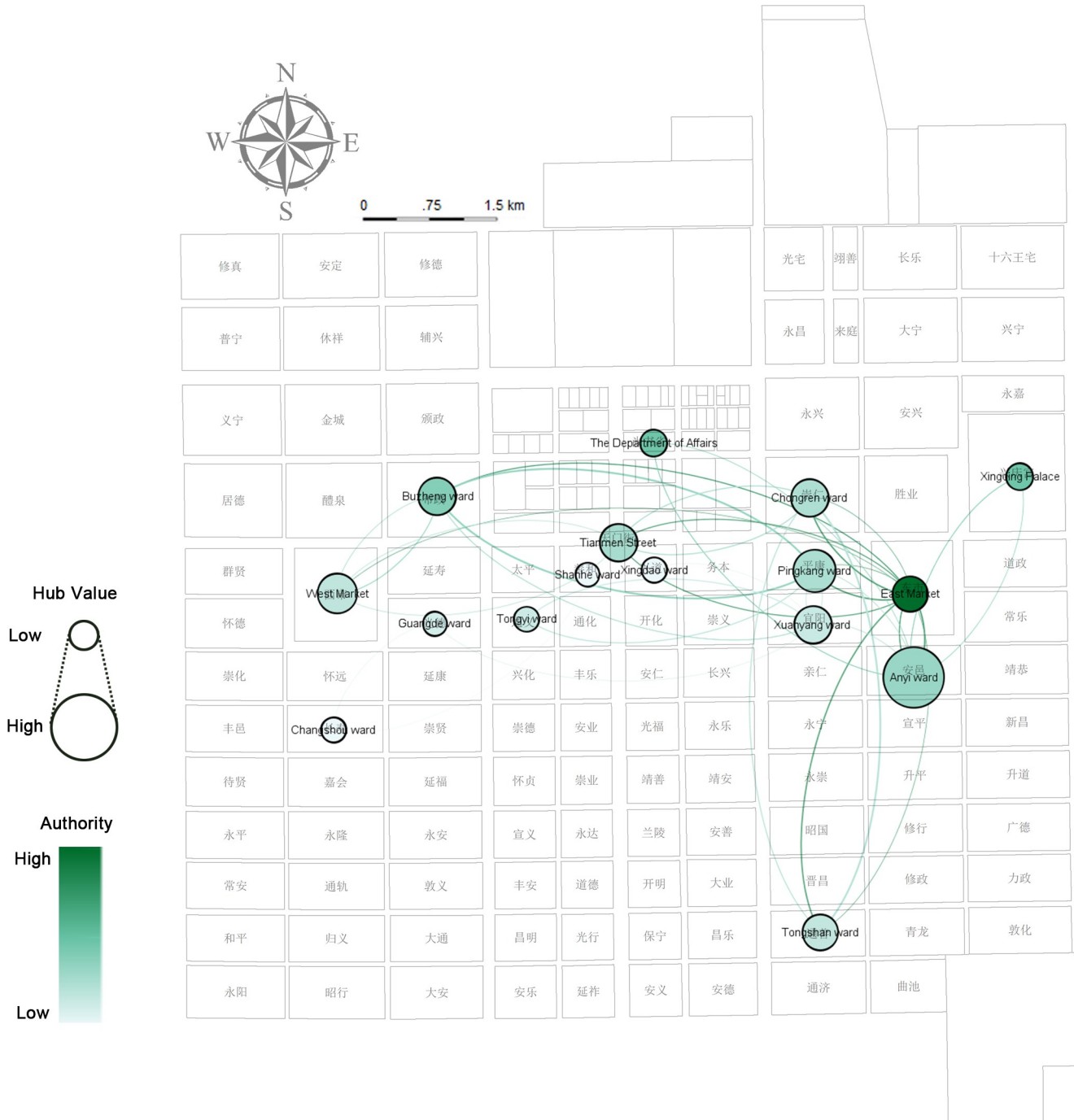

**Fig 21. Network analysis of characters and places in authority and hub analysis based on a full-text, spatially embedded, directed network of characters.**

Chang'an for a long time (Fig 22, S3 Table). Thus, this dramatic story may create the resonance of early growth that belongs to contemporary literati members such as Bai.

This feasible digital spatial–temporal narrative model combines the text with space and the literary narrative with historical narrative to gain a novel and unique view of "what-how-why" about this classical novel's narrative experience based on the experience of the researchers who

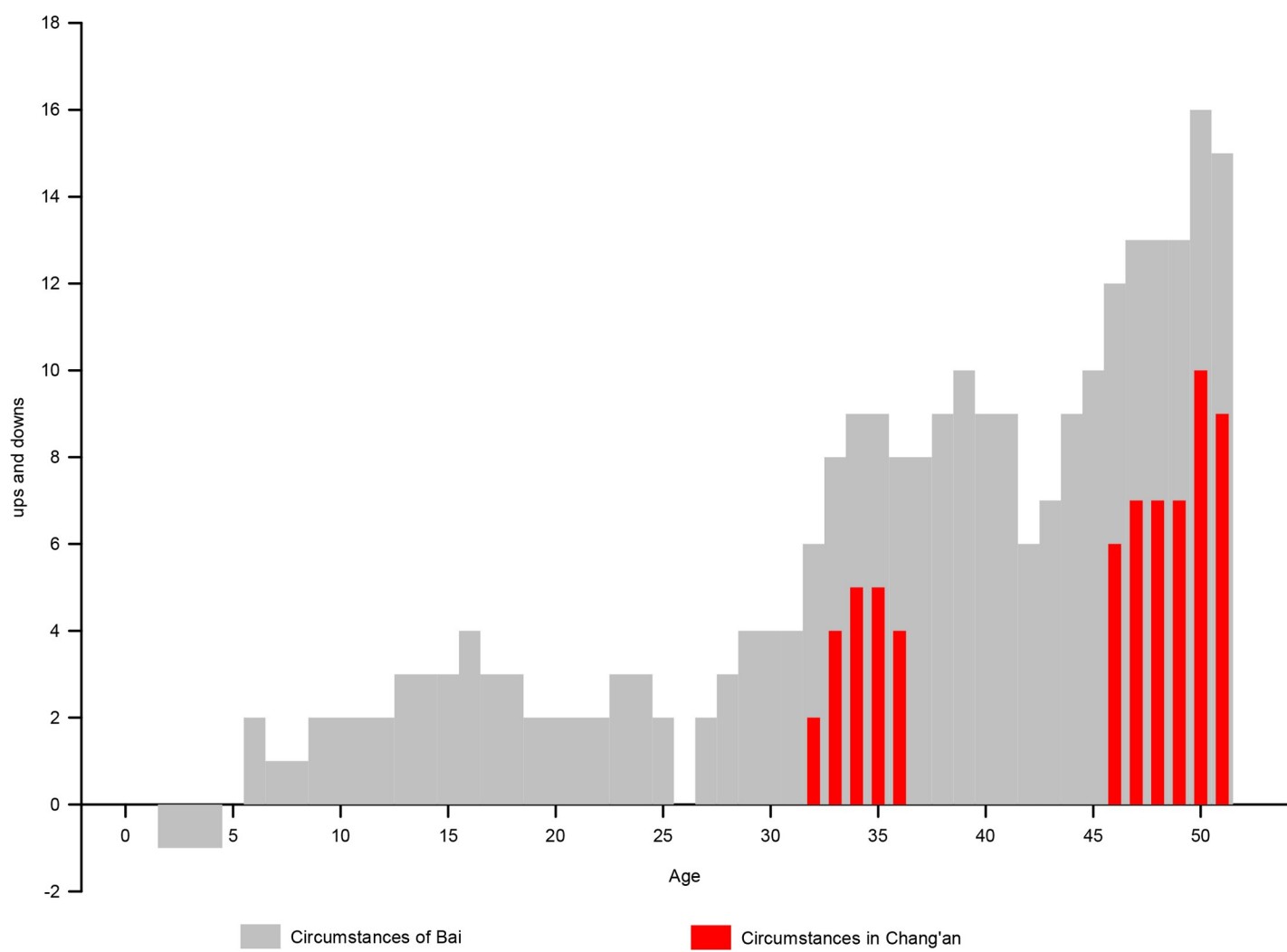

**Fig 22. Bai Xingjian's life circumstances.** From the age of 32–36 years, when he entered the bureaucracy and was promoted and then released; from the age of 46 to his death, when he resided in Chang'an and was gradually promoted [32].

can be considered as non-professional literature researchers (all majored in architecture, urban planning, or landscape architecture). In future studies, additional thematic reading experiences should be investigated and organized in the knowledge graph of this novel, including the narrative of combined linear time and juxtaposed space.

The representative value of Tang Tales cannot be examined merely through the limited methodology presented in this paper. This transparent and dialogic framework still has considerable potential for its application, including applicable objects and the corresponding interpretation of the representation. To this end, starting from *The Tale of Li Wa*, at least a real interactive platform based on digital reconstructions of all Tang Tales as interconnected intermediaries, allowing for psychological tests, should be established to improve the relevant comprehension of Tang Chang'an's empirical knowledge and its narratives. In an ongoing study by the authors on over four hundred Tang Tales, the spatially-embedded network of characters demonstrates that the literati have both a sense of self and a political consensus on Chang'an. This collective imagination demonstrates a similar pattern to that of *The Tale of Li Wa*, proving the latter to be, once again, the perfect experimental object.

## Supporting information

**S1 File. Scanned version.**
(PDF)

**S2 File. Raster map of Tang Chang'an with location information.**
(ZIP)

**S3 File. Temporal narrative in space.**
(ZIP)

**S4 File. Space syntax of Chang'an roads.**
(GRAPH)

**S5 File. Temporal simulation path in space.**
(ZIP)

**S6 File. Places' various statistical value.**
(ZIP)

**S7 File. Characters' co-occurrence network.**
(ZIP)

**S8 File. Spatially-embedded network of characters.**
(ZIP)

**S1 Table. Word level.**
(XLSX)

**S2 Table. Phrase level.**
(XLSX)

**S3 Table. Chronicle of Bai Xingjian.**
(XLSX)

## Acknowledgments

The authors gratefully acknowledge the Tang Chang'an GIS basemap provided by Prof. Pan Wei from Yunan University and his team from the GIS Lab at Northwest Institute of Historical Environment and Socio-Economic Development of Shaanxi Normal University. The first version of the historical GIS data of Tang Chang'an in this research is from Prof. Timothy Baker of National Dong Hwa University and Dr. Liao Hsiung-Ming of Academia Sinica. Mr. Xiao Tianyi from School of Architecture of Tianjin University contributes to the training of a word-vector model of the texts via Python.

## Author Contributions

**Conceptualization:** Zhaoyi Ma, Jie He, Shuaishuai Liu.

**Data curation:** Zhaoyi Ma, Shuaishuai Liu.

**Formal analysis:** Zhaoyi Ma, Shuaishuai Liu.

**Funding acquisition:** Jie He.

**Investigation:** Zhaoyi Ma, Shuaishuai Liu.

**Methodology:** Zhaoyi Ma, Jie He.

**Project administration:** Jie He.

**Resources:** Jie He.

**Software:** Zhaoyi Ma, Shuaishuai Liu.

**Supervision:** Zhaoyi Ma, Jie He.

**Validation:** Zhaoyi Ma, Shuaishuai Liu.

**Visualization:** Zhaoyi Ma, Shuaishuai Liu.

**Writing – original draft:** Zhaoyi Ma.

**Writing – review & editing:** Zhaoyi Ma, Jie He.

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
