## [Decision Letter · Decision Letter 0]

8 Jul 2019

PONE-D-19-16990

Representation of the spatio-temporal narrative of The Tale of Li Wa李娃传

PLOS ONE

Dear Dr. He,

Thank you for submitting your manuscript to PLOS ONE. After careful consideration, we feel that it has merit but does not fully meet PLOS ONE’s publication criteria as it currently stands. Therefore, we invite you to submit a revised version of the manuscript that addresses the points raised during the review process.

Both reviews (I also agree with them) pointed out the lack of research question in the current version. Please reshape the revised manuscript accordingly. In addition, please make sure to address the other issues raised by the reviewers in the revisions.

We would appreciate receiving your revised manuscript by Aug 22 2019 11:59PM. To enhance the reproducibility of your results, we recommend that if applicable you deposit your laboratory protocols in protocols.io, where a protocol can be assigned its own identifier (DOI) such that it can be cited independently in the future. For instructions see: http://journals.plos.org/plosone/s/submission-guidelines#loc-laboratory-protocols

We look forward to receiving your revised manuscript.

Kind regards,

Hyejin Youn

Academic Editor

PLOS ONE

Journal Requirements:

2.  In order to meet the requirements for the Science of Stories collection, the Guest Editors ask that you please make the code to reproduce your analysis available in a stable, public repository (for example, Zenodo, or GitHub) or a suitable cloud computing service (such as Code Ocean) when submitting your revised manuscript. The code should include a license file and detailed readme so that someone with access to the dataset is able to reproduce your analysis using the code. We ask that you include the DOI for the repository holding your code in an updated Data Availability statement with your revised manuscript.

3. Please ensure that you include a title page within your main document. You should list all authors and all affiliations as per our author instructions and clearly indicate the corresponding author.

Reviewers' comments:

Reviewer's Responses to Questions

**Comments to the Author**

1. Is the manuscript technically sound, and do the data support the conclusions?

Reviewer #1: Yes

Reviewer #2: Partly

2. Has the statistical analysis been performed appropriately and rigorously? 

Reviewer #1: Yes

Reviewer #2: N/A

3. Have the authors made all data underlying the findings in their manuscript fully available?

Reviewer #1: Yes

Reviewer #2: Yes

4. Is the manuscript presented in an intelligible fashion and written in standard English?

Reviewer #1: Yes

Reviewer #2: No

5. Review Comments to the Author

Reviewer #1: The purpose of this article is to open a dialogue between science and literature, and attempt a diachronic and synchronic approach to a a Chinese classic novel, The Tale of Li Wu.

The article refers to three concepts:

- The chronotope, which is a philological notion proposed by the literature theorist Mikhail Bakhtin ; the chronotope covers the spatial and temporal description elements contained in a fictional or non-fictional narrative and considers place and time as being in solidarity.

- Gabriel Zoran, for which the text refers the reader to a certain model of external reality by means of which he must reconstruct the world.

- W.J.T. Mitchell, for whom we cannot think about literature or anything else without using spatial metaphors ; according to this author, "the search for and momentary imposition of spatial patterns on the temporal flow of literature is a central aspect of reading"

Through various statistics (about sentiment, POS, characters and places), several points were studied: the narrative skills, the cityscape of Chang'an and the structural relationship between places, the life of the residents of Chang'an and the relations between them, and the theme of the story. The authors propose many statistical analyses to make a "total" reading of the novel. However, it is not so much a question of understanding as of visualizing the text.

Introduction :

The authors seem to summarize well the main research questions about this novel. However, I could not check the validity of many references for problems of linguistic understanding (there are in chinese). Readers therefore need to begin with an act of trust.

Note that readers may wish to have a short summary of statistical approaches to literature (without going into details). It would be possible, for example, to quote Franco Moretti's book, Graphs Maps Tree (among others).

Analyses :

The whole paper is well constructed and the statistics seem solid (as far as I can estimated them). Some minor revisions could be required :

- It is easy to interpret graphs and diagrams, even from formal statistics, yet it is difficult to verify the veracity of these interpretations. It’s not to say, of course, that the interpretation of some graphs is false here, but it would be better to indicate somewhere this call for caution.

- The word "chronotope" comes up 13 times and deserves to be explained at a little more length for readers who do not know Bakhtin.

- I have no doubt that the text is magnificent (and the authors made me want to read it), but the terms "superb" "splendid" or "shocking" does not sound very scientific. The analysis has nothing to do with the quality of the novel.

- Do the networks based on a full-text spatially embedded undirected network of characters correspond to small world properties identified by Duncan J. Watts and Steven H. Strogatz (1998) (i.e. small average shortest path length close to purely random graphs and large clustering coefficient significantly higher than expected by random chance) ? if so, what does that tell us about the novel?

- Bakhtin is sometimes misspelled, as well as chronotope. As I am not a native English speaker, I cannot comment on spelling in general.

To conclude, I took a real pleasure in reading this paper, which I feel is well deserved to be published with minor revisions.

Reviewer #2: In this manuscript, Representation of the patio-temporal narrative of The Tale of Li Wa, Ma et al. provided a rather systematic study of the spatial and temporal structure of a traditional Chinese novel, the tale of Li Wa. To help modern readers to understand this classic, the authors have adopted several techniques and methods from data mining and network science. Although I find the study systematic and somewhat interesting, I have several questions regarding the “research question” of the paper and some technical details of the methods.

First, despite several methods have been adopted and hundreds of analysis have been made, it is very difficult to find a very clear statement of the research question in the paper. Is it about a demonstration of a new method? (It seems not, since all methods have been invented before). Or is it about using these analysis to support a known theory? Or the authors want to present some new insights from the analysis? The authors may have a clear answer to these questions and may have some deep insights in their mind, but it is very difficult (at least for me) to find the answer in the current presentation of the work. The current version is so over overwhelmed with different analysis without a clear thread to link them. Without a key message, it is also hard for the readers to tell the purpose of each analysis and understand the relationship between them. For example, what is the message the author want to convey in Fig. 17? What is the new insight of the plot? How could that deepen the reader’s understanding of the book since it is like a confirmation that the main character is Zheng? By the way, is the color representing the communities learned from the network, did you use the same method shown in Fig. 21? If a different modularity method is adopted (say Modularity Q proposed by Girvan and Newman), whether the results shown in these two figures still hold?

In this sense, I suggest the author reorganize the narrative and structure of the paper, highlighting the research question and novelty of the paper. Rather than solely demonstrating the toolbox, I hope the author could provide the insight of each analysis.

Second, I also have a couple of comments about some technical details.

a) The assignment of the sentiment orientation (so) value seems a little bit arbitrary, since several words and characters could have ambiguous meaning and it is possible for different people have different feelings about the same word. Could the authors provide some cross-validations of the value assignment to make sure the value indeed present the basic expression? For example, maybe the authors could borrow tools from network science (as they did in the paper), and build a network of the sentiment expressions? (if two sentiment expressions are shown in one sentence or from one reading time snapshot, a link will be added between them) Of course, the author could try other alternative methods to do the cross-validation.

b) Can the authors provide the details of how do they measure the “stratum” in Fig. 18? Why the compass also have a color of light blue? Does that mean the person has shown there? The same issue for the lower panel of Fig. 13.

6. PLOS authors have the option to publish the peer review history of their article (what does this mean?). If published, this will include your full peer review and any attached files.

Reviewer #1: No

Reviewer #2: No

---

## [Author Response · Author response to Decision Letter 0]

10 Sep 2019

1. Response to the “lack of research” question pointed out by the editor

Editor’s comment: Both reviews (I also agree with them) pointed out the lack of research question in the current version. Please reshape the revised manuscript accordingly. In addition, please make sure to address the other issues raised by the reviewers in the revisions.

Thanks for the editor's suggestion. We therefore rethought our research question and reorganized the entire paper with modifications. We propose a framework of digital models that integrate literary theory, providing contemporary readers with an additional opportunity to understand Chinese classical novels wrote in an almost unfamiliar language.

We show one of these possibilities under this framework through a knowledge model or narrative based on our team’s experience. It is the the gradually in-depth theme of "what-how-why" about the growth of the male protagonist hidden in the novel narritives. This knowledge model is gradually obtained through in-depth analysis and concatenation of various diagrams in the "Time-Space-time-Space" logical loop proposed in the manuscript.

1.1 Outline of the revised manuscript

People living in the digital age seem to have more difficulties when reading classical novels, in terms of obscure words, contextual difference and reading habits. This paper proposes a framework of digital models integrating literary theory to represent the narrative of a Chinese classic novel, The Tale of Li Wa, which has been diversely favored by literature and historians in the past approximately 900 years. To imitate reader’s reading psychological changes, we refer to three abstract spatial narrative theories: Bakhtin’s chronotopos (time–space) (Bakntin, 2002), which enlightens us to add historical context to the model; Zoran’s topographical-chronotopic-textual space (Zoran, 1984), which inspires us to reconstruct reader's reading experience in three dimensions; and Mitchell’s spatial metaphors (Mitchell, 1980), which indicates the core of reading lies in pursuit and acquisition of the changing pattern and meaning in the process of reading. In terms of methods, quantitative analysis including semantic analysis, sentiment analysis, spatial analysis, and network analysis are applied. Through the collation, analysis, and transmedia argumentation of information, normal readers can have the opportunity to choose their own model to represent and interpret the texts in their own perspective in the text’s knowledge graph. This paper presents one of these possible themes on illustrating the growth of male protagonist in our open framework of "Time-Space-time-Space", which opens up dialogues between computation and literature, diachronic and synchronic, reader and the author.

The time level shows the basic variables’ trajectories of read time, as well as the point of the story which is a growth story of the male protagonist when he first entering the capital Chang'an before servicing the bureaucracies. The space-time level unfolds various variables’ trajectories in the context of a two-dimensional urban space of read time, as well as the story’s details about how he lives and grows in the complex urban space of Chang'an. The space level represents the statistical pattern of mapping trajectories of different variables and a meaningful system indicating the social space of Chang'an City described in the novel. Through the integration of the chronotope of author, text, and reader, the understanding on the narrative, the physical background of Chang’an city, and the life of the residents of Chang’an are gradually profound within the looped spatio-temporal framework. We even find the primitive image of power control embodied in the story. From a different perspective, it helps us to have a better understanding on the author Bai Xingjian's bureaucratic career and his experience in Chang'an in the author's chronotope.

Our team explores the representation of spatio-temporal narrative of classical literature from a perspective of non-professional literature researchers (all majored in architecture, urban planning, and landscape architecture), and tries to gradually and opaquely interpret The Tale of Li Wa based on researchers’ personal experience. We must admit that this project is not a digital humanities project based entirely on specific content or assumptions. But such a process also proves the feasibility of our digital framework model from the perspective of reader's reading experience, which combines history with texts, the diachronic with synchronic. This research can be considered as a comprehensive exploration towards “narratives, experiences and geographical spaces”.

2. Responses to suggestion of minor revisions from Reviewer #1

Many thanks to Reviewer #1 for your deep understanding and appreciation of our paper.

2.1 Questions about References

Reviewer’s comment: The authors seem to summarize well the main research questions about this novel. However, I could not check the validity of many references for problems of linguistic understanding (there are in chinese). Readers therefore need to begin with an act of trust.

Note that readers may wish to have a short summary of statistical approaches to literature (without going into details). It would be possible, for example, to quote Franco Moretti's book, Graphs Maps Tree (among others).

The electronic version of all references except the web references have been uploaded on github (https://github.com/aayi/The-Tale-of-Li-Wa). In view of some previous studies on The tale of Li Wa, I recommend that readers read its entry on Wikipedia（https://en.m.wikipedia.org/wiki/The_Tale_of_Li_Wa）. Actually there are a lot of previous literature about this novel, and I only select some typical ones to discuss in our paper. In addition, we supplemented our novelty based on these literature review.

More reference in Chinese can be found basically on CNKI (China National Knowledge Infrastructure, http://new.oversea.cnki.net/index/). Table 1 listed their url which may at least provide bilingual abstracts. 

Table 1. Information About Chinese Reference.

Title URL DOI

[Cultural Interpretation of Chinese Narrative Literature] 中国叙事学的文化阐释 http://cn.oversea.cnki.net/kcms/detail/detail.aspx?DbCode=CJFR&dbname=CJFD2003&filename=GDMZ200303007&uid=WEEvREcwSlJHSldRa1FhdkJkVG1EWmxjVTFCT3ZkL1hKUGkvOVJBNk4ybz0=$9A4hF_YAuvQ5obgVAqNKPCYcEjKensW4IQMovwHtwkF4VYPoHbKxJw!!

[History of Chinese Literature: Volumn on Sui]中国文学史话:隋唐五代卷 http://book.kongfz.com/162251/728624782/

[Review of the Biography of Li Wa Research in the Past Ten Years] 近十年《李娃传》研究述评 http://new.oversea.cnki.net/KCMS/detail/detail.aspx?dbcode=CJFQ&dbname=CJFDLAST2017&filename=LNSW201701003&uid=WEEvREcwSlJHSldRa1FhdkJkVG1EWmxjVTFCT3ZkL1hKUGkvOVJBNk4ybz0=$9A4hF_YAuvQ5obgVAqNKPCYcEjKensW4IQMovwHtwkF4VYPoHbKxJw!!&v=Mjk1ODc0SDliTXJvOUZaNFI4ZVgxTHV4WVM3RGgxVDNxVHJXTTFGckNVUkxPZmIrUnZGeW5uVmJyT0tTUFllYkc=

10.13888/j.cnki.jsie(ss).2017.01.003

[On the New Field of Narratology: space narrotology] 空间叙事学：叙事学研究的新领域 http://new.oversea.cnki.net/KCMS/detail/detail.aspx?dbcode=CJFQ&dbname=CJFD2008&filename=TJSS200806010&uid=WEEvREcwSlJHSldRa1FhdkJkVG1EWmxjVTFCT3ZkL1hKUGkvOVJBNk4ybz0=$9A4hF_YAuvQ5obgVAqNKPCYcEjKensW4IQMovwHtwkF4VYPoHbKxJw!!&v=MDc3NjIxTHV4WVM3RGgxVDNxVHJXTTFGckNVUkxPZmIrUnZGeW5nVkx6UE1TZllmYkc0SHRuTXFZOUVaSVI4ZVg=

[Spatial Turn in Narrative Theories-A Summary of Spatial Theories in Narrative] 叙事理论的空间转向——叙事空间理论概述 http://new.oversea.cnki.net/KCMS/detail/detail.aspx?dbcode=CJFQ&dbname=CJFD2007&filename=JXSH200711008&uid=WEEvREcwSlJHSldRa1FhdkJkVG1EWmxjVTFCT3ZkL1hKUGkvOVJBNk4ybz0=$9A4hF_YAuvQ5obgVAqNKPCYcEjKensW4IQMovwHtwkF4VYPoHbKxJw!!&v=MDAwNDFyQ1VSTE9mYitSdkZ5bmdXcnpQTHpYWVpyRzRIdGJOcm85RmJJUjhlWDFMdXhZUzdEaDFUM3FUcldNMUY=

[Shuli Criticism on the Plots of "The Story of Li Wa"] 《李娃传》情节数理批评 http://cn.oversea.cnki.net/kcms/detail/detail.aspx?DbCode=CJFR&dbname=CJFD2013&filename=GLJY201303017&uid=WEEvREcwSlJHSldRa1FhdkJkVG1EWmxjVTFCT3ZkL1hKUGkvOVJBNk4ybz0=$9A4hF_YAuvQ5obgVAqNKPCYcEjKensW4IQMovwHtwkF4VYPoHbKxJw!!

[Research on the Evolution of The Tale of Li Wa] 《李娃传》嬗变研究 http://new.oversea.cnki.net/KCMS/detail/detail.aspx?dbcode=CJFQ&dbname=CJFD9495&filename=NJDX403.017&uid=WEEvREcwSlJHSldRa1FhdkJkVG1EWmxjVTFCT3ZkL1hKUGkvOVJBNk4ybz0=$9A4hF_YAuvQ5obgVAqNKPCYcEjKensW4IQMovwHtwkF4VYPoHbKxJw!!&v=Mjc0ODZlNEhjL01yb2dxRjU0T2ZnZzV6aEFVNGpoNE9YNlRySDAzZWJPZFJiS2RaK2R2RXlua1VRPT1LeWZQZHI=

[Chang'an and Tales in Late Tang Dynasty: Focusing on the Analysis of The Tale of Li Wa] 唐代后期的长安与传奇小说——以《李娃传》的分析为中心 https://book.douban.com/subject/1064289/

[Automatic Word Segmention of Middle Ancient Chinese Texts with CRFs] 基于CRFs和词典信息的中古汉语自动分词 http://new.oversea.cnki.net/KCMS/detail/detail.aspx?dbcode=CJFQ&dbname=CJFDLAST2017&filename=XDTQ201705011&uid=WEEvREcwSlJHSldRa1FhdkJkVG1EWmxjVTFCT3ZkL1hKUGkvOVJBNk4ybz0=$9A4hF_YAuvQ5obgVAqNKPCYcEjKensW4IQMovwHtwkF4VYPoHbKxJw!!&v=MDAwNDFyQ1VSTE9mYitSdkZ5bmhVNy9KUFNuZmY3RzRIOWJNcW85RVpZUjhlWDFMdXhZUzdEaDFUM3FUcldNMUY=

[Standard of POS Tag of Contemporary Chinese for CIP (revise)] 信息处理用现代汉语词类标记规范（修订） http://new.oversea.cnki.net/KCMS/detail/detail.aspx?dbcode=CPFD&dbname=CPFD9908&filename=JBYW200512001052&uid=WEEvREcwSlJHSldRa1FhdkJkVG1EWmxjVTFCT3ZkL1hKUGkvOVJBNk4ybz0=$9A4hF_YAuvQ5obgVAqNKPCYcEjKensW4IQMovwHtwkF4VYPoHbKxJw!!&v=MDU5NDY3TUkxNFRMeS9TZWJHNEh0VE5yWTlGWmVzS0RoTkt1aGRobmo5OFRuanFxeGRFZU1PVUtyaWZadTl2SGl2a1U3

[Dictionary of Tang Chang’an] 唐代长安词典 https://book.douban.com/subject/4205523/

[A chronicle of Bai Xingjian] 白行简年谱 http://new.oversea.cnki.net/KCMS/detail/detail.aspx?dbcode=CJFQ&dbname=CJFD2002&filename=WNXI200203005&uid=WEEvREcwSlJHSldRa1FhdkJkVG1EWmxjVTFCT3ZkL1hKUGkvOVJBNk4ybz0=$9A4hF_YAuvQ5obgVAqNKPCYcEjKensW4IQMovwHtwkF4VYPoHbKxJw!!&v=MDAwNDFyQ1VSTE9mYitSdkZ5bmhVTHZQTWlQVFo3RzRIdFBNckk5RllZUjhlWDFMdXhZUzdEaDFUM3FUcldNMUY=

2.2 Suggestion for cautiously interpreting graphs

Reviewer’s comment: It is easy to interpret graphs and diagrams, even from formal statistics, yet it is difficult to verify the veracity of these interpretations. It’s not to say, of course, that the interpretation of some graphs is false here, but it would be better to indicate somewhere this call for caution.

Visualization is powerful, especially for contemporary readers who are used to reading pictures. For this reason in the first version of our manuscript, we used different kinds of visual graphs and diagrams to show one of the possible interpretations by reading the digital representation of texts. However, your valuable suggestion reminds us that the figures are more accurate. So we re-examined the visualized graphs and replaced one with table. Specific values of some parameters are also indicted to in the manuscript to explain the rest figures. 

2.3 Suggestion for the summary of statistical approaches to literature

Reviewer’s comment: Note that readers may wish to have a short summary of statistical approaches to literature (without going into details). It would be possible, for example, to quote Franco Moretti's book, Graphs Maps Tree (among others).

A new review conclusion of the past researches has been added as “Most of these mentioned academic studies are not much easier to read than the classical novel itself. And because of the limitations of perspective and the opacity of personal feelings, it is difficult for readers to synthesize their own experience from previous research.”

A description has been added in the Introduction parts is as follows: “Most of the modern readers are in the digital age when the dialogue between humanities data and computation moves from text digitization, text structuring to text semantics. The representational and interpretive practices to literary texts through quantitative analysis and transmedia argumentation can be traced in Franco Moretti’s ideas about ‘distant reading’ in 2000.”

2.4 Suggestion for the explanation of chronotope

Reviewer’s comment: The word "chronotope" comes up 13 times and deserves to be explained at a little more length for readers who do not know Bakhtin.

Thank you for the reminding. We have given a brief explanation as “This term was adapted by Bakhtin from Einstein's theory of relativity, meaning a spatio-temporal literary unity”

2.5 Suggestion for deleting unscientific terms

Reviewer’s comment: I have no doubt that the text is magnificent (and the authors made me want to read it), but the terms "superb" "splendid" or "shocking" does not sound very scientific. The analysis has nothing to do with the quality of the novel.

Thank you for the suggestion. We have deleted these terms or substituted with more academic expressions.

2.6 Questions about the overall structure of the network in network analysis

Reviewer’s comment: Do the networks based on a full-text spatially embedded undirected network of characters correspond to small world properties identified by Duncan J. Watts and Steven H. Strogatz (1998) (i.e. small average shortest path length close to purely random graphs and large clustering coefficient significantly higher than expected by random chance) ? if so, what does that tell us about the novel?

Based on your suggestion, we have listed the parameter files for network analysis in the paper. Our analysis of the overall characteristics of the two network analyses is as following: 

1. The characters’ co-occurrence network is a scale-free network, which may prove that the whole story revolves around the male protagonist.

2. The spatially embedded undirected network of characters is a stable small world network, which may prove the setting stages have intentional elements rather than random nor regular.

However, the judgment of network parameters in literature is still a metaphorical category, and it is still expected that more experts will join.

2.7 Questions about the English spelling

Reviewer’s comment: Bakhtin is sometimes misspelled, as well as chronotope. As I am not a native English speaker, I cannot comment on spelling in general.

Thank you for reminding us and we have revised the spelling mistakes.

3. Responses to suggestion of reorganizing the paper from Reviewer #2

Thank the reviewers#2 for your suggestions on this paper, which inspires us to rethink the whole structure of the article.

3.1 About research question and paper’s narrative

Reviewer’s comment: In this sense, I suggest the author reorganize the narrative and structure of the paper, highlighting the research question and novelty of the paper. Rather than solely demonstrating the toolbox, I hope the author could provide the insight of each analysis.

 I have several questions regarding the “research question” of the paper and some technical details of the methods.

First, despite several methods have been adopted and hundreds of analysis have been made, it is very difficult to find a very clear statement of the research question in the paper. Is it about a demonstration of a new method? (It seems not, since all methods have been invented before). Or is it about using these analysis to support a known theory? Or the authors want to present some new insights from the analysis? The authors may have a clear answer to these questions and may have some deep insights in their mind, but it is very difficult (at least for me) to find the answer in the current presentation of the work. The current version is so over overwhelmed with different analysis without a clear thread to link them. Without a key message, it is also hard for the readers to tell the purpose of each analysis and understand the relationship between them. …

In this sense, I suggest the author reorganize the narrative and structure of the paper, highlighting the research question and novelty of the paper. Rather than solely demonstrating the toolbox, I hope the author could provide the insight of each analysis.

We had hoped to organize a framework of sets of digital models that integrate literary theory, thus providing contemporary readers with an additional opportunity to understand Chinese classical novels more appropriately. But the spatio-temporal representation in the last manuscript did not pay much attention to the construction of the overall logic. Actually we were a bit cautious about reorganizing the narrative from the author's perspective in the last version since we wanted to provide readers with the freedom to read in-depth, which however made it difficult to read our paper.

Your suggestion is very valuable. In this revised manuscript, we decided to show one of the possibilities under this framework of sets of digital models that integrate literary theory. It is a knowledge model or narrative to illustrate the theme "what-how-why" about the growth of the male protagonist based on the our team’s experience. This knowledge model is gradually obtained through in-depth analysis (we have made more in-depth interpretation of diagrams and there do exist some new insights which we choose not to highlight) and concatenation of various diagrams in a logical loop of "Time-Space-time-Space".

3.2 About different community detection algorithms

Reviewer’s comment: For example, what is the message the author want to convey in Fig. 17? What is the new insight of the plot? How could that deepen the reader’s understanding of the book since it is like a confirmation that the main character is Zheng? By the way, is the color representing the communities learned from the network, did you use the same method shown in Fig. 21? If a different modularity method is adopted (say Modularity Q proposed by Girvan and Newman), whether the results shown in these two figures still hold?

Different algorithms may indeed lead to different answers, especially for community detection algorithm of network analysis, because communities have relatively vague boundaries. However, this modularity analysis at the "space" level is relatively in line with the more close analysis in the "time" and "space-time" level. Therefore we think it has some kind of credibility. But we do not deny the interpretation nature of this digital representation, which is a problem we will not or not able to avoid. We just show a possibility.

3.3 About the assignment of the sentiment orientation (so) value

Reviewer’s comment: The assignment of the sentiment orientation (so) value seems a little bit arbitrary, since several words and characters could have ambiguous meaning and it is possible for different people have different feelings about the same word. Could the authors provide some cross-validations of the value assignment to make sure the value indeed present the basic expression? For example, maybe the authors could borrow tools from network science (as they did in the paper), and build a network of the sentiment expressions? (if two sentiment expressions are shown in one sentence or from one reading time snapshot, a link will be added between them) Of course, the author could try other alternative methods to do the cross-validation.

We actually have done two rounds of SO assignment. The percentage of consent of two rounds of SO value assignment is >0.9.

3.4 Questions about non-standard cartography

Reviewer’s comment: Why the compass also have a color of light blue? Does that mean the person has shown there? The same issue for the lower panel of Fig. 13.

We have narrowed the boundary of IDW interpolation analysis.

3.5 Questions about details of measuring the “stratum”

Reviewer’s comment: Can the authors provide the details of how do they measure the “stratum” in Fig. 18?

We replace the interpolation analysis of social stratum with a statistical chart, and count the social stratum of characters in each place, weighted by the number of times characters enter places. The classification of social stratum in the story from untouchable to nobles is as follows: (1) level 1: beggar, servant, and sex worker; (2) level 2: businessman, civilian, madam; (3) level 3: ward head (里胥), candidate student, successful candidate, county judicial official (贼曹); and (4) level 4: Chang’an officials (京尹) and officials from other places.

References:

Bakhtin MM. Forms of Time and of the Chronotope in the Novel: Notes toward a Historical Poetics. In: Richardson B, editors. Narrative dynamics: Essays on time, plot, closure, and frames. Columbus: The Ohio University; 2002. pp. 15-24.

Zoran G. Towards a Theory of Space in Narrative. Poetics Today. 1984; 5(2): 309-335.

Mitchell WJT. Spatial Form in Literature: Toward a General Theory. Critical Inquiry. 1980; 6(3): 539-567.

---

## [Decision Letter · Decision Letter 1]

21 Oct 2019

PONE-D-19-16990R1

Representation of the spatio-temporal narrative of The Tale of Li Wa李娃传

PLOS ONE

Dear Dr. He,

Thank you for submitting your manuscript to PLOS ONE. After careful consideration, we feel that it has merit but does not fully meet PLOS ONE’s publication criteria as it currently stands. Therefore, we invite you to submit a revised version of the manuscript that addresses the points raised during the review process. 

I agree with Reviewer 1 regarding writing and presentation style of the manuscript. There are a few places that can be better presented. Please resubmit after addressing the issues raised by Reviewer 2. 

We would appreciate receiving your revised manuscript by Dec 05 2019 11:59PM. To enhance the reproducibility of your results, we recommend that if applicable you deposit your laboratory protocols in protocols.io, where a protocol can be assigned its own identifier (DOI) such that it can be cited independently in the future. For instructions see: http://journals.plos.org/plosone/s/submission-guidelines#loc-laboratory-protocols

We look forward to receiving your revised manuscript.

Kind regards,

Hyejin Youn

Academic Editor

PLOS ONE

Reviewers' comments:

Reviewer's Responses to Questions

**Comments to the Author**

1. If the authors have adequately addressed your comments raised in a previous round of review and you feel that this manuscript is now acceptable for publication, you may indicate that here to bypass the “Comments to the Author” section, enter your conflict of interest statement in the “Confidential to Editor” section, and submit your "Accept" recommendation.

Reviewer #1: All comments have been addressed

Reviewer #2: (No Response)

2. Is the manuscript technically sound, and do the data support the conclusions?

Reviewer #1: Yes

Reviewer #2: Partly

3. Has the statistical analysis been performed appropriately and rigorously? 

Reviewer #1: Yes

Reviewer #2: N/A

4. Have the authors made all data underlying the findings in their manuscript fully available?

Reviewer #1: Yes

Reviewer #2: Yes

5. Is the manuscript presented in an intelligible fashion and written in standard English?

Reviewer #1: No

Reviewer #2: Yes

6. Review Comments to the Author

Reviewer #1: In this form, the paper seems to be much more understandable and better constructed.

A new reference that might be of interest (?): In his analysis of the book L'Éducation sentimentale (in La Distinction. Critique sociale du jugement, 1979), Bourdieu used a map of the Latin Quarter (i.e. Quartier Latin, in Paris) and gives a relational reading: he shows the places of geographical (and therefore social) origin of the different protagonists and makes it possible to point out positions that take on a large surplus of meaning when we oppose / put them in relationship with each other.

NOTE: About question 5, not being a native English speaker, I cannot judge the spelling of the paper.

Reviewer #2: I am glad to see the revised version of the paper Representation of the patio-temporal narrative of The Tale of Li Wa by Ma et al. I appreciate that the authors have made several changes to their manuscript, especially their endeavor in providing a more organized presentation of the work. Although I find the paper is largely improved, I still expect some more detailed answers to some of my previous questions.

1) About the community detection method (Answer 3.2). I am still expecting some explanations of the insights of the figure. Did it change our previous understanding of the narrative structure of the work? Or it only confirms that Zheng is the main character?

And I still recommend the authors to try other algorithms of community detection (e.g. Modularity Q, etc.). I fully understand that here the author only presents one possibility, and results obtained from other algorithms could be very different. But the authors should report the differences in their paper and let the reader know there ARE other possibilities and only one possibility is presented here.

2) About the sentiment orientation (SO) value (Answer 3.3). The authors state that they have done two rounds of SO assignment, which I fully appreciate. But this is more like a robustness test of the convergence of the method, instead of a validation of the method itself. I am still wondering that, like what I said in the previous comments, did the SO value indeed present the basic expression? Could the authors present more detailed information or empirical evidence of this point?

3) I also have a new question about the revised manuscript. In line 532, the authors added that the network is “a small-sized scale-free network”, they said “a few nodes known as the hubs that have extremely many connections/edges”. But it seems to me that the largest node in the network only has a degree around 10 to 20, which makes it very hard to be a scale-free network. Did I miss something? Did the authors do some statistics test about this statement?

By answering these lingering questions, in my opinion, the authors may provide a clearer presentation of the work.

7. PLOS authors have the option to publish the peer review history of their article (what does this mean?). If published, this will include your full peer review and any attached files.

Reviewer #1: No

Reviewer #2: No

---

## [Author Response · Author response to Decision Letter 1]

5 Dec 2019

Responses to Reviewers

1. Response to the “writing and presentation style” issue noted by the editor

Editor’s comment: I agree with Reviewer 1 regarding writing and presentation style of the manuscript. There are a few places that can be better presented.

We have revised the writing and presentation style of the manuscript to the best of our capability and the manuscript has revised by a professional language editor. Further, we have explained and revised some technical details of the methods, including the addition of another community detection algorithm, supplementary statistics regarding emotion-related linguistic rules, and an explanation of the network’s degree distribution.

2. Responses to the suggestion of algorithms and analysis from Reviewer #1

Thank you for your suggestions regarding the technical details of the methods, which have inspired us to reexamine the methods and make some necessary revisions.

2.1 About community detection method

Reviewer’s comment: About the community detection method (Answer 3.2). I am still expecting some explanations of the insights of the figure. Did it change our previous understanding of the narrative structure of the work? Or it only confirms that Zheng is the main character?

And I still recommend the authors to try other algorithms of community detection (e.g. Modularity Q, etc.). I fully understand that here the author only presents one possibility, and results obtained from other algorithms could be very different. But the authors should report the differences in their paper and let the reader know there ARE other possibilities and only one possibility is presented here.

Because the characters’ co-occurrence network is not a large-scale network, based on your recommendation, it seemed advisable to attempt other divisive algorithms such as Girvan–Newman (Girvan and Newman, 2002) for a hierarchical community structure rather than the agglomerative algorithm alone (they are both based on modularity). In general, there is no difference between the two algorithms, although the former is slightly more accurate because it involves a top-down approach. Nonetheless, our data show an interesting phenomenon: the conflict of characters in the top-down model (Fig 1 below) seems to be much weaker than that in the bottom-up model (Fig 2), although the overall structure of the two is similar. According to Fig 1, Zheng’s father and Zheng are from the same class, whereas Fig 2 shows Zheng’s embarrassing situation between his father and Li Wa, which is somewhat similar to the different views proposed by scholars with different perspectives. However, it should be noted that the co-occurrence network of characters is not a real social network in the novel. In actuality, many characters around Zheng such as the predecessor have potential social relations with others. We have included further discussion concerning the topic in our manuscript in lines 538-547.

Fig 1. Statistics of characters in co-occurrence network, cluster and betweenness centrality in a. new Girvan–Newman divisive algorithm for community detection

Fig 2. Statistics of characters in co-occurrence network, cluster and betweenness centrality in original agglomerative algorithm for community detection, use edge weights off

We also tested the spatially embedded network of characters. The results of top-down model (Fig 3) seem to be a slightly more complex to explain from the perspective of literary geography than the bottom-up model (Fig 4), although there are many similarities between the overall structures of the two. Therefore, we decided not to include the new model’s results in the manuscript, as this may require a considerable amount of sources or data for a comparative study of literary-geographical phenomena.

Fig 3. Network analysis of characters and places in Girvan–Newman clustering analysis and weighted degree centrality based on a full-text, spatially embedded, undirected network of characters.

Fig 4. Network analysis of characters and places in modularity class analysis (use edge weights on) and weighted degree centrality based on a full-text, spatially embedded, undirected network of characters.

2.2 About the assignment of the sentiment orientation (SO) value

Reviewer’s comment: About the sentiment orientation (SO) value (Answer 3.3). The authors state that they have done two rounds of SO assignment, which I fully appreciate. But this is more like a robustness test of the convergence of the method, instead of a validation of the method itself. I am still wondering that, like what I said in the previous comments, did the SO value indeed present the basic expression? Could the authors present more detailed information or empirical evidence of this point?

Thank you for your suggestion on sentiment analysis. We have included further discussion concerning the topic in our manuscript in lines 179-184. We take the algorithm of “document sentiment classification using sentiment lexicons” (Liu, 2015) to calculate the emotional curve (the trajectory of the integral function of SO value) in the framework of time and space. In terms of the sentiment lexicon generation, there are three main approaches: a manual approach, a dictionary-based approach, and a corpus-based approach. However, the latter two methods are based on the first, which exploit some linguistic rules or conventions based on manually-collected seed sentiment words to identify other sentiment words (Liu, 2015). Because there are no extant and ready-made sentiment lexicons, we use manual assignment for the sentiment lexicon. The artificial test of emotion curve regarding the plot’s ups and downs in our manuscript also proves that our assignment and the algorithm that is related to sentiment have some advantages.

As mentioned in the manuscript (in Word-level analysis), the linguistic rules of ancient Chinese are considerable different from those of modern Chinese, and we cannot determine the rules that should be used here to measure the accuracy of the manual assignment of SO. Nevertheless, based on your suggestion, we have made an interesting attempt, assuming that the sentiment words in ancient Chinese follow the same logic for current sentiment analyses in that sentiment words have context-dependent orientations, i.e., the total distance among words with the same orientation sentiment expression is closer than that among different ones (Turney et al., 2003, Kamps et al., 2004).

Based on the unigrams of The Tale of Li Wa removing stop words, we complete the training of its word2vec model with gensim package (parameters: sg = 0, size = 50, window = 5, min_count = 1, iter = 20) and acquire the correlation (cosine_similarity, −0.4 ~ 1) between each two words––the closer the value is to 1, the higher the correlation (based on vector distance) between them (Rehurek et al., 2010). Consequently, we take this correlation between word A (Source) and word B (Target) multiplied by the SO value of word B as the weight of the edge, establishing a sentiment network of words in Gephi. Gephi’s calculated weighted outdegree of each word reflects its relationship with all other sentiment words, i.e., the larger the weighted outdegree a word is, the more positive its sentiment is, and vice versa. However, the results of different type of manually-collected sentient words (1, −1, 0) reveal that the ancient Chinese may not simply conform to the hypothesis that the sentiment words have context-dependent orientations (Fig 5.a–c), as our SO assignment is not so outrageous. This part has been simply added to the manuscript in the part of “Word-level analysis”, as well as to my Github project (https://github.com/aayi/The-Tale-of-Li-Wa).

Fig 5. Weighted outdegree distribution of words’ sentiment network (a. SO = 1, b. SO = −1, c. SO = 0).

2.3 About the scale-free network

Reviewer’s comment: In line 532, the authors added that the network is “a small-sized scale-free network”, they said “a few nodes known as the hubs that have extremely many connections/edges”. But it seems to me that the largest node in the network only has a degree around 10 to 20, which makes it very hard to be a scale-free network. Did I miss something? Did the authors do some statistics test about this statement?

Figure 6. Degree distribution. Figure 7. Degree distribution (ln-ln).

We have reconfirmed the characters’ co-occurrence network whose degree distribution turns out to asymptotically follow a power law (Figs 6–7). It is very likely a power-law network with scale-free property, although it is not a large-scale network (汪小帆et al., 2012, 郭世泽et al., 2012). However, to be rigorous, we decided to revise the phrasing in the manuscript (lines 532–533) to “a probable small-scale power-law network.”

3. Responses to Reviewer #2

Thank you for your suggestions regarding our paper.

3.1 About the new reference

Reviewer’s comment: A new reference that might be of interest (?): In his analysis of the book L'Éducation sentimentale (in La Distinction. Critique sociale du jugement, 1979), Bourdieu used a map of the Latin Quarter (i.e. Quartier Latin, in Paris) and gives a relational reading: he shows the places of geographical (and therefore social) origin of the different protagonists and makes it possible to point out positions that take on a large surplus of meaning when we oppose / put them in relationship with each other.

This new reference has reminded us to include further supplements to the field of the literary cartography as a quantitative and transmedia approach to literary geography in our manuscript–– “…can be traced in Franco Moretti’s … and his practice of literary cartography integrating “spatial criticism” in 1999” (lines 125–126).

3.2 Questions on spelling

Reviewer’s comment: About question 5, not being a native English speaker, I cannot judge the spelling of the paper.

Thank you for the reminder; we have made revisions within the manuscript with respect to language and grammar to improve accuracy and readability.

References

Girvan M, Newman MEJ. Community structure in social and biological networks. Proceedings of the National Academy of Sciences. 2002 Jun; 99(12): 7821-7826. doi: 10.1073/pnas.122653799.

Kamps J, Marx M, Mokken RJ, et al. Using WordNet to measure semantic orientation of adjectives. In: Maria TL, Maria FX, Fátima F, Rute C, Raquel S, editors. Proceedings of the Fourth International Conference on Language Resources and Evaluation; 2004 May; Lisbon: European Language Resources Association, 2004. pp.1115-1118.

Liu B. Sentiment analysis: Mining opinions, sentiments, and emotions. New York: Cambridge University; 2015.

Rehurek R, Sojka P. Software framework for topic modelling with large corpora. In: Rene W, Hamish C, Jon P, Elena B, Ekaterina B, Udo H, Karin V, Anni RC, editors. Proceedings of the LREC 2010 Workshop on New Challenges for NLP Frameworks; 2010 May 22; Valletta: ELRA; 2010. pp. 45-50.

Turney PD, Micharel LL. Measuring praise and criticism: Inference of semantic orientation from association. ACM Trans Inf Syst. 2003; 21(4): 315-46.

汪小帆, 李翔, 陈关荣. [Network science: An introduction] 网络科学导论. Beijing: Higher Education Press; 2012. Chinese.

郭世泽，陆哲明. [The basic theory of complex network] 复杂网络基础理论. Beijing: Science Press; 2012. Chinese.

---

## [Decision Letter · Decision Letter 2]

14 Feb 2020

PONE-D-19-16990R2

Representation of the spatio-temporal narrative of The Tale of Li Wa李娃传

PLOS ONE

Dear Dr. He,

Thank you for submitting your manuscript to PLOS ONE. After careful consideration, we feel that it has merit but does not fully meet PLOS ONE’s publication criteria as it currently stands. Therefore, we invite you to submit a revised version of the manuscript that addresses the last comment raised during the review process.

I agree with the reviewer's comment about the claim of having analysed a "small scale free network".  If the authors believe it is a very relevant point to discuss and to support, they can attempt more advanced interpolation techniques as suggested by the reviewer, an submit an improved version of the fit, which however in my opinion would probably make very little sense anyhow, see 

https://science.sciencemag.org/content/335/6069/665

My suggestion here is just to avoid specifying "scale free network" in the paper, if possible, limiting the statement to something on the line of "a few nodes known as the hubs that have extremely many connections/edges". If this correction is promptly done, I would be able to accept the paper without further review.

We would appreciate receiving your revised manuscript by Mar 30 2020 11:59PM. To enhance the reproducibility of your results, we recommend that if applicable you deposit your laboratory protocols in protocols.io, where a protocol can be assigned its own identifier (DOI) such that it can be cited independently in the future. For instructions see: http://journals.plos.org/plosone/s/submission-guidelines#loc-laboratory-protocols

We look forward to receiving your revised manuscript.

Kind regards,

Riccardo Gallotti

Academic Editor

PLOS ONE

Reviewers' comments:

Reviewer's Responses to Questions

**Comments to the Author**

1. If the authors have adequately addressed your comments raised in a previous round of review and you feel that this manuscript is now acceptable for publication, you may indicate that here to bypass the “Comments to the Author” section, enter your conflict of interest statement in the “Confidential to Editor” section, and submit your "Accept" recommendation.

Reviewer #2: (No Response)

2. Is the manuscript technically sound, and do the data support the conclusions?

Reviewer #2: Partly

3. Has the statistical analysis been performed appropriately and rigorously? 

Reviewer #2: N/A

4. Have the authors made all data underlying the findings in their manuscript fully available?

Reviewer #2: Yes

5. Is the manuscript presented in an intelligible fashion and written in standard English?

Reviewer #2: Yes

6. Review Comments to the Author

Reviewer #2: The authors have answered most of my questions and I appreciate very much that the authors have made several changes to the paper. I only have one concern left for point 2.3. Based on the degree distribution plot they showed, I am still not quite convinced that it is a scale-free network, or "Power-law network" as they phrased. I suggest the author to do one more statistical test following the paper "power law distributions in empirical data" by Aaron Clauset et al., where they have provided a rather accurate way to test a distribution is power law or not. Their codes are public available online.

7. PLOS authors have the option to publish the peer review history of their article (what does this mean?). If published, this will include your full peer review and any attached files.

Reviewer #2: No

---

## [Author Response · Author response to Decision Letter 2]

24 Mar 2020

1. Response to the “small scale free network” issue noted by the editor and the reviewer

Editor’s comment: I agree with the reviewer's comment about the claim of having analysed a "small scale free network". If the authors believe it is a very relevant point to discuss and to support, they can attempt more advanced interpolation techniques as suggested by the reviewer, an submit an improved version of the fit, which however in my opinion would probably make very little sense anyhow, see https://science.sciencemag.org/content/335/6069/665

My suggestion here is just to avoid specifying "scale free network" in the paper, if possible, limiting the statement to something on the line of "a few nodes known as the hubs that have extremely many connections/edges". If this correction is promptly done, I would be able to accept the paper without further review.

Reviewer’s comment: Based on the degree distribution plot they showed, I am still not quite convinced that it is a scale-free network, or "Power-law network" as they phrased. I suggest the author to do one more statistical test following the paper "power law distributions in empirical data" by Aaron Clauset et al., where they have provided a rather accurate way to test a distribution is power law or not. Their codes are public available online.

Thanks for the reviewer’s comment and the editor’s advice. We have followed the editor's advice and revised the manuscript as follows: “Character statistics enable a network analysis of the characters’ co-occurrence network in the entire document in Fig 17 and 18 (see S7 File), which indicates a network around hubs (a few nodes have much more connections/edges).” (lines 531–533)

2. Responses to the laboratory protocols from the editor

Thank you for your suggestions, we have included the DOI link (http://dx.doi.org/10.17504/protocols.io.bdyni7ve) in the Materials and methods section of our manuscript.

3. Other problems

We have also revised other minor errors in the manuscript.

---

## [Decision Letter · Decision Letter 3]

26 Mar 2020

Representation of the spatio-temporal narrative of The Tale of Li Wa李娃传

PONE-D-19-16990R3

Dear Dr. He,

We are pleased to inform you that your manuscript has been judged scientifically suitable for publication and will be formally accepted for publication once it complies with all outstanding technical requirements.

With kind regards,

Riccardo Gallotti

Academic Editor

PLOS ONE

Additional Editor Comments (optional):

Reviewers' comments:

Reviewer's Responses to Questions

**Comments to the Author**

1. If the authors have adequately addressed your comments raised in a previous round of review and you feel that this manuscript is now acceptable for publication, you may indicate that here to bypass the “Comments to the Author” section, enter your conflict of interest statement in the “Confidential to Editor” section, and submit your "Accept" recommendation.

Reviewer #2: All comments have been addressed

2. Is the manuscript technically sound, and do the data support the conclusions?

Reviewer #2: Yes

3. Has the statistical analysis been performed appropriately and rigorously? 

Reviewer #2: Yes

4. Have the authors made all data underlying the findings in their manuscript fully available?

Reviewer #2: Yes

5. Is the manuscript presented in an intelligible fashion and written in standard English?

Reviewer #2: Yes

6. Review Comments to the Author

Reviewer #2: (No Response)

7. PLOS authors have the option to publish the peer review history of their article (what does this mean?). If published, this will include your full peer review and any attached files.

Reviewer #2: No

---

## [Editor Report · Acceptance letter]

1 Apr 2020

PONE-D-19-16990R3 

Representation of the spatio-temporal narrative of The Tale of Li Wa李娃传 

Dear Dr. He:

I am pleased to inform you that your manuscript has been deemed suitable for publication in PLOS ONE. Congratulations! Your manuscript is now with our production department. 

With kind regards,

on behalf of

Dr. Riccardo Gallotti 

Academic Editor

PLOS ONE